



# Optimizing Rainfall-Triggered Landslide Thresholds to Warning Daily Landslide Hazard in Three Gorges Reservoir Area

Bo Peng[1], Xueling Wu[1]

[1]School of Geophysics and Geomatics, China University of Geosciences, Wuhan 430074, China

*Correspondence to*: Xueling Wu (wuxl@cug.edu.cn)

**Abstract.** Rainfall is intrinsically connected to the incidence of landslide catastrophes. Exploring the ideal rainfall threshold model (RTM) for an area in order to determine the rainfall warning level (RWL) for the region for daily landslide hazard warning (LHW) is critical for precise prevention and management of local landslides. In this paper, a method for calculating rainfall thresholds using multilayer perceptron (MLP) regression is proposed for 453 rainfall-induced landslides. First, the

study area was divided into subareas based on topography and climate conditions. Then, two methods, MLP and ordinary least squares (OLS), were utilized to explore the optimal RTM for each subregion. Subsequently, 11 factors along with three models were selected to predict landslide susceptibility (LS). Finally, to obtain daily LHW result for the study area, a superposition matrix was employed to overlay the daily RWL with the ideal LS prediction results. The following are the study's findings: (1) The optimal RTMs and calculation methods are different for different subregions. (2) The Three-

dimensional convolutional neural network model produces more accurate LS prediction results. (3) The daily LHW was validated using anticipated rainfall data for July 19, 2020, and the validation results proved the correctness of the LHW results and RTM.

## 1 Introduction

Landslide catastrophes accounted for 71.55% of geological disasters in China from 2005 to 2021, according to the China

Statistical Yearbook (http://www.stats.gov.cn/sj/ndsj/). Frequent landslide catastrophes endanger people's lives and property (Xing et al., 2021). Rainfall will lead to landslide disasters by changing the pore pressure in the soil body (Zhao et al., 2022) and weakening the shear strength of the geotechnical body (Chan et al., 2018). According to research (Marin et al., 2020; Yuniawan et al., 2022): rainfall is intrinsically connected to the great majority of landslide deformation and instability. Therefore, it is significant to delineate the rainfall thresholds for different rainfall conditions and areas through the study for

the fine development of landslide hazard warning (LHW) and disaster prevention and control. LHW is described as the conditional prediction of probable landslide temporal and spatial probability under the limitations of triggering and inducing variables (Budimir et al., 2015). The rainfall warning level (RWL) (i.e., the temporal probability of landslide occurrence) calculated by the rainfall threshold model (RTM) is the triggering factor in this study, and the inducing factor is the



prediction result of landslide susceptibility (LS) (i.e., the spatial probability of landslide occurrence) calculated by the
susceptibility model.

The spatial probability of landslide occurrence can be reflected by LS (Huang et al., 2022b). General linear models (Aksha et al., 2020), information value models (Yu et al., 2022), machine learning models, and others are among the methods used to predict LS. Machine learning models can fit and predict the nonlinear relationship between LS and landslide predisposing factors more effectively than other kinds of models (Guo et al., 2021). Commonly used machine learning models include
logistic regression (Baharvand et al., 2020), artificial neural networks (Jiang et al., 2014), support vector machines (SVM) (Zhu and Hu, 2012), random forests (RF) (Chen et al., 2014), Bayesian algorithms (He et al., 2019) and deep learning algorithms (Huang et al., 2020). However, determining which model is best suited for LS prediction is challenging, and there is great uncertainty in the LS prediction results of various machine learning models (Xia et al., 2020). Even little improvements in LS prediction accuracy might have a significant influence on LS zoning (Chen et al., 2018). Therefore, to
decrease the uncertainty of LS results, different susceptibility models are frequently employed to predict LS in the study area, and the model with the greatest accuracy is chosen.

RTM approaches primarily include of deterministic methods based on physical and hydrological models, as well as empirical methods based on landslide cataloguing and rainfall event statistics (Chung et al., 2017; Wu et al., 2015). The former establishes the relationship between rainfall and landslide stability through dynamic hydrological models and
determines the critical rainfall threshold for landslide instability in the physical model (Ciurleo et al., 2019). However, due to the difficulty in accurately obtaining hydrological parameters and geotechnical parameters on a large scale, this method is only applicable to smaller study area (Wu and Yeh, 2020). The latter is mainly derived by calculating the relationship between historical landslide and rainfall data (Abraham et al., 2020a; Pradhan et al., 2019). This method is widely used because of its advantages of convenience in data acquisition, strong applicability, and excellent results (Martinovic et al.,
2018). Currently, commonly used RTM include the intensity of rainfall - duration of rainfall (I-D) threshold model (Abraham et al., 2019; Lee et al., 2014) and effective rainfall - duration of rainfall (E-D) threshold model (Abraham et al., 2020b; Peruccacci et al., 2017). The regression methods used to calculate the RTM include logistic regression (Mathew et al., 2014), ordinary least squares (OLS) regression (Rossi et al., 2017) and quantile regression (Salee et al., 2022). There are differences in the applicability of different RTM and different regression methods in different regions (Marin, 2020; Segoni
et al., 2018). Therefore, to decrease uncertainty in LHW, several regression methods and RTM must be used to establish the best appropriate rainfall threshold for a certain location.

Given that many researchers have employed the log-log coordinates system for RTM regression analysis (He et al., 2020), this study proposes to use MLP regression method to study the rainfall thresholds under various rainfall durations. Simultaneously, the third dimension indicator "rainfall for the day" (R) was introduced to create the E-D-R RTM based on
the E-D RTM (Liu et al., 2022).





In this study, the Three Gorges Reservoir Area (TGRA) was used as the study area, and the landslides were first catalogued to get the E and D data during the five days before the landslides, as well as the R data at the time of the landslides. Following that, the rainfall thresholds corresponding to the E-D and E-D-R models for distinct landslide occurrence probabilities were calculated using both OLS and MLP regression methods, respectively. To explore the optimal RTM for
the study area and the feasibility of neural network for RTM research, as well as to categorize RWL based on the optimal RTM. Then, select the factors that induce landslide occurrence and predict the LS results using RF, SVM, and 3D convolutional neural network (CNN-3D) models, and utilize the LS results with the best accuracy as the spatial probability of landslide occurrence in the study area. Finally, the daily RWL is combined with the LS result using the superposition matrix to achieve the daily LHW, which serves as a reference for precision prevention and management of local landslide
disasters.

## 2. Methods

### 2.1 Rainfall Threshold Model

#### 2.1.1 OLS Regression

OLS regression is a commonly used linear regression method that can be used to establish a linear relationship between the
independent variable ($x$) and the dependent variable ($y$). It minimizes the error between the predicted value and the actual observed value by seeking the slope and intercept that best fits the data (Lim et al., 2023).

The basic form of its regression model can be expressed as:

$$y = \beta_0 + \sum_{i=1}^{n} \beta_i x_i \,, \tag{1}$$

where $y$ denotes the dependent variable, $x_i$ denotes the independent variable, $n$ denotes the number of independent variables,
$\beta_i$ denotes the coefficients of the independent variables, and $\beta_0$ denotes the constant intercept.

#### 2.1.2 MLP Regression

MLP is a common neural network with the ability of nonlinear mapping, which can learn complex nonlinear functional relationships through multiple layers of nodes. Currently, it has been widely used in many fields such as geospatial analysis (Hasan et al., 2023; Wang et al., 2023b), aerodynamics (Barcenas et al., 2023), atmospheric science (Hoffman and Jasinski,
2023), rainfall prediction (Narimani et al., 2023), and image fusion (Mei et al., 2023). In the regression analysis of scatter data, a scatter data set can be regarded as composed of multiple input-output data pairs, and the model adjusts the weights of the model by minimizing the error between the predicted value and the actual data, and finally realizes the regression of scatter data. In this study, we built an MLP model with two hidden layers (Fig. 1).





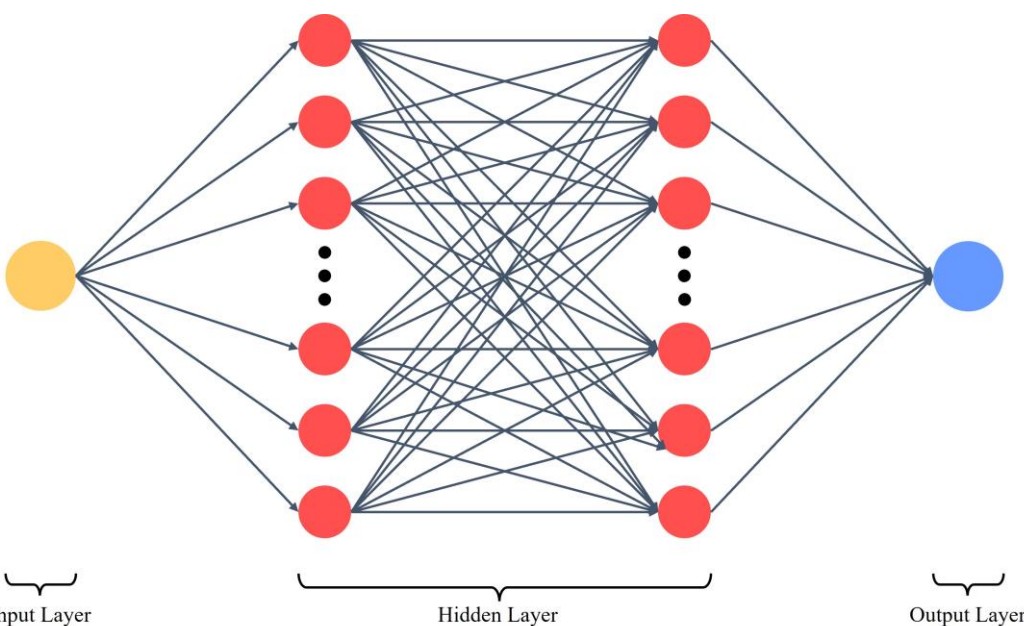

**Figure 1: Schematic diagram of the MLP model.**

### 2.1.3 E-D-R Rainfall Threshold Model

The E-D-R RTM is based on the E-D RTM, with the introduction of the R metrics at the third latitude to optimize the original RTM. To investigate the E-D-R RTM, the E-D RTM must first be determined.

The E-D RTM aims to investigate the effective rainfall as a function of duration of rainfall (Teja et al., 2019). The scatter is
generally analyzed by regression in a log-log coordinates system, and then the resulting fitted straight line is transformed into a result in a Cartesian coordinate system. The expression for this is:

$$E = \alpha \times D^{\beta} , \tag{2}$$

Assume that the linear equation obtained by fitting in the log-log coordinates system has an intercept of $b$ and a slope of $a$. Then, in the above equation, $\alpha = 10^{b}$, $\beta = a$, and $D$ denotes the duration of rainfall (d). E is the effective rainfall (mm),
which refers to the total amount of rainfall that actually infiltrates and acts on the landslide body in addition to the slope runoff and evaporation (Huang et al., 2022a). The effective rainfall formula used in this study is as follows:

$$E = \sum_{i=1}^{n} k^{i-1} E_i , \tag{3}$$

where $E$ denotes the effective rainfall, $E_i$ is the rainfall on the previous $i$ days, and $k$ is the effective rainfall coefficient. The value of k is usually set to 0.8 (Huang et al., 2022a). Furthermore, it has been shown that the effective rainfall in the first 5
days of the TGRA has a strong link with landslide events (Zhou et al., 2022). Therefore, the number of days of rainfall statistics $n$ in this work is set to 5.





The third dimension of the indicator R is added based on the E-D RTM to expand the threshold model from two to three dimensions, and the RTL meet the following relational equation:

$$T = \max\{G_E, G_R\},\tag{4}$$

where $T$ denotes the final RWL, while $G_E$ and $G_R$ denote the RWL for the E-D model and R, respectively.

## 2.2 CNN-3D Model

Convolutional Neural Network (CNN) is a deep learning algorithm, widely used in image recognition (Fan et al., 2022; Gill et al., 2022), natural language processing (Jin et al., 2023; Kaliyar et al., 2021) and other domains. Its primary concept is to extract features from input data using a convolution operation (Youssef et al., 2022). However, for one- and two-dimensional

CNNs, feature extraction for induced factor data is only performed at a single raster point. Both methods ignore the spatial information around the raster points (Yang et al., 2022). As a result, this study presents CNN-3D in order to fully use the rich spatial information around the raster points in order to increase the prediction accuracy of LS. The structure of CNN-3D is similar to that of CNN, but since the input data contains more information, CNN-3D can provide more accurate results (Liu et al., 2023).

We picked a three-dimensional structure to create samples in this experiment. Before producing the samples, an n-channel picture is formed by superimposing n components. Each pixel is then extended outwards by 7 pixels to generate a $15 \times 15 \times$ n image as input. Subsequently, through operations such as convolution and pooling in the hidden layer, the high-level features are mapped to the low-dimensional space and stored in the neural units of the fully connected layer, and finally classified using the Softmax function to obtain the results of landslides and non-slides (Fig. 2).

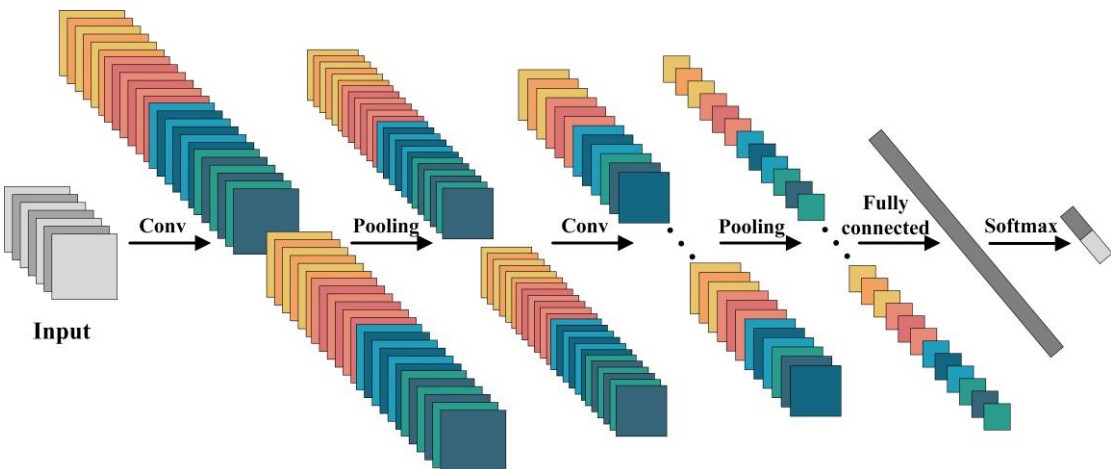


**Figure 2: Schematic diagram of CNN-3D structure.**





## 3. Overview of the Study Area

### 3.1 Physical and Geographical Characteristics

The study area is located in the upper reaches of the Yangtze River between Sandouping in Yichang City and Jiangjin
District in Chongqing, which is situated at longitude 105°50′- 111°42′ E and latitude 28°30′-31°45′ N (Cheng et al., 2022), encompassing a total of 29 administrative districts and counties in Hubei Province and Chongqing Municipality (7 districts and counties in Hubei, and 22 districts and counties in Chongqing), and covering a total area of $5.67 \times 10^4 km^2$ (Fig. 3).The climate of the region is subtropical monsoon with an average annual precipitation of 445-1813 mm (Long et al., 2021). And the abundant rainfall in the area is a major factor inducing landslides (Guo et al., 2022).

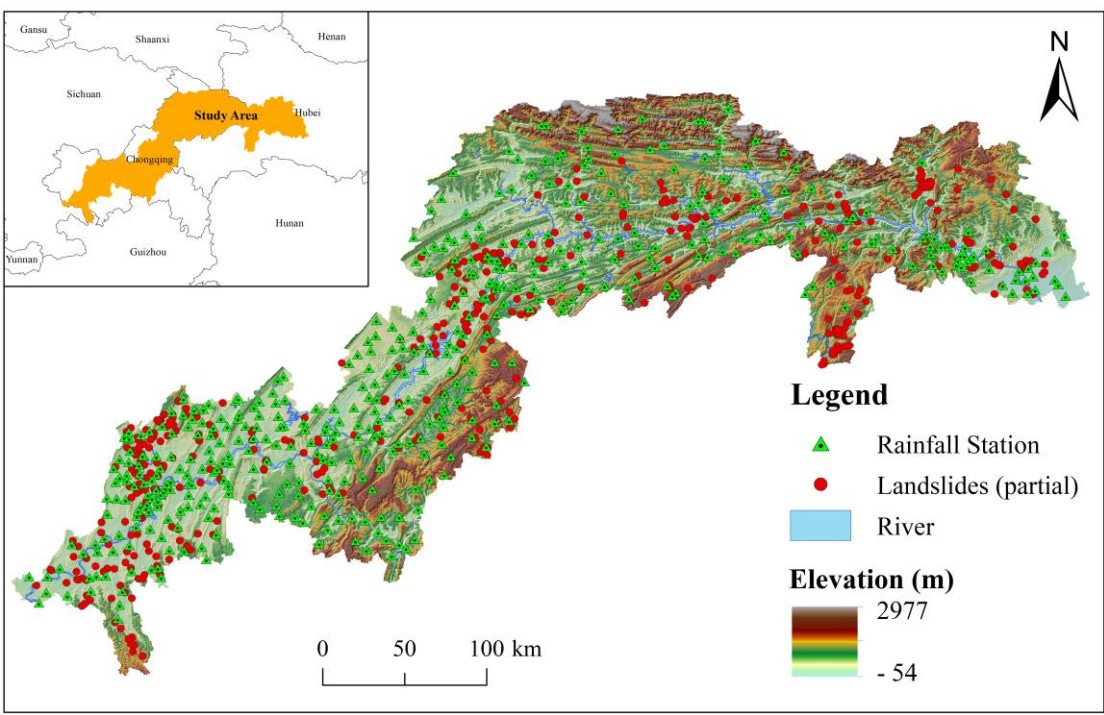


**Figure 3: Geographic location of study area.**

### 3.2 Study Area Subdivision and Landslide Data Cataloguing

Geomorphology, geology, and climate play the most important role in preparatory process of landslide initiation in any region (Dahal and Hasegawa, 2008), and the differences between them lead to different rainfall thresholds in various regions.
Therefore, in this study, the whole study area was divided into 10 zones (Fig. 4) by considering the topography and climatic conditions of the study area, and the optimal RTM was calculated for each zone separately. Among them, $Z_{11}$, $Z_{12}$ and $Z_{13}$ are the moderate rainfall zone, low rainfall zone and high rainfall zone in the folded region; $Z_{21}$, $Z_{22}$, $Z_{23}$, $Z_{24}$ and $Z_{25}$ are the low rainfall zone, relatively high rainfall zone, high rainfall zone, moderate rainfall zone and high rainfall zone in the low





and medium mountain region, respectively; $Z_3$ is the high rainfall zone in the medium and high mountain region; and $Z_4$ is

the high rainfall zone in the hilly and plain zone.

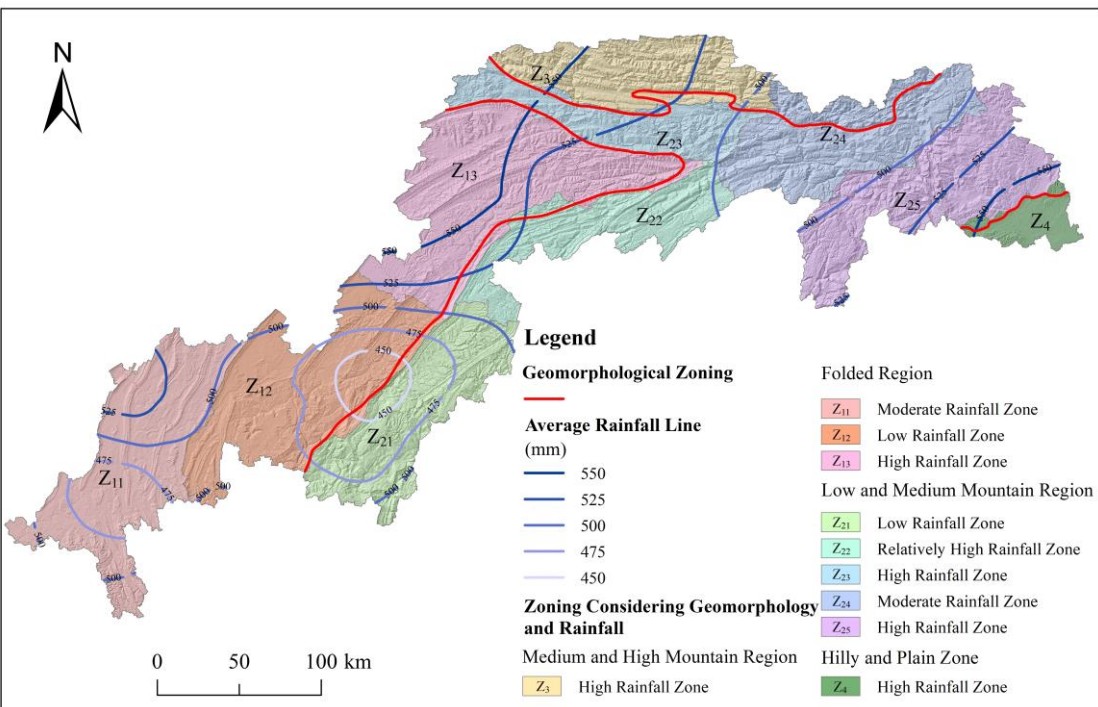

**Figure 4: Zoning map of the study area.**

Before landslide data cataloguing, the corresponding rainfall dataset needs to be acquired. Based on the abundance of rainfall

stations in the study area (refer to Fig. 3, Rainfall Station), Thiessen polygon method were used for the delineation (Zhao et

al., 2019), which facilitates the finding of rainfall stations corresponding to landslides. The Thiessen polygon method results

satisfy the following conditions: (1) each polygon contains one and only one rainfall station; (2) any point within each

polygon is the closest to the rainfall station within the unit; (3) the points on the boundary are the same distance to the two

neighboring rainfall stations. The result of its division is shown in Fig. 5.

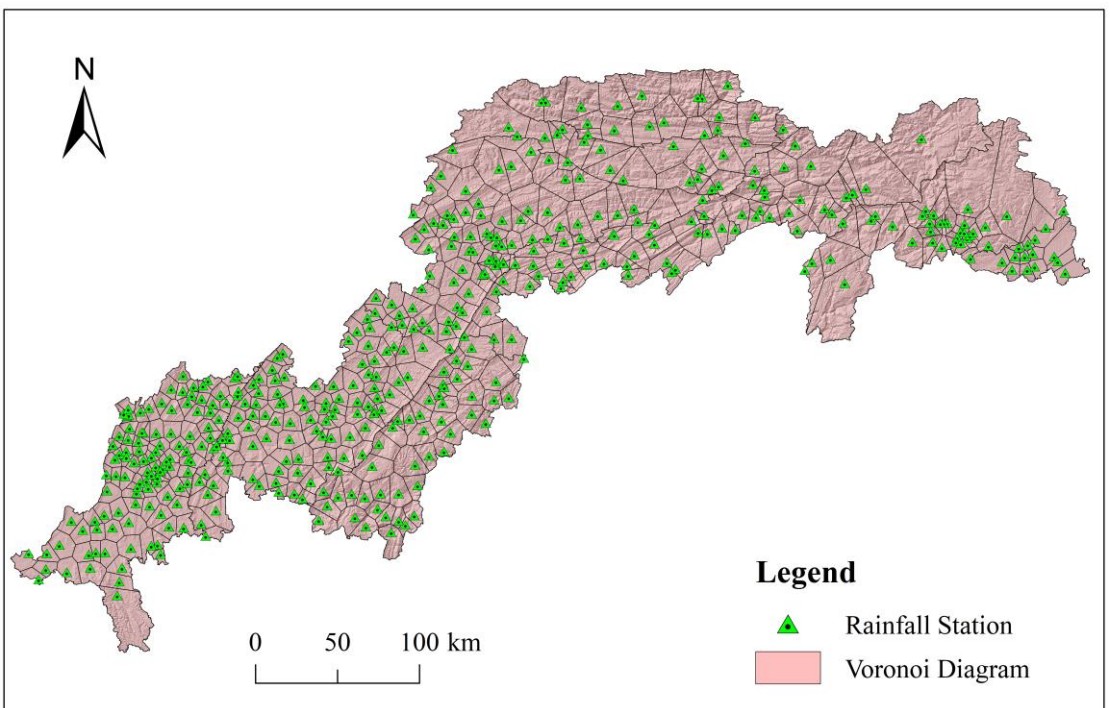

**Figure 5: Thiessen polygon method results map.**

Landslide data cataloguing is the basis for the study of rainfall thresholds (Gariano et al., 2021), and its main contents include basic information such as the time of occurrence of landslides, geographic location, associated rainfall stations, and so on. The landslide cataloguing data in this study were obtained from the historical landslide hazard data provided by Wuhan Geological Survey Centre (http://www.wuhan.cgs.gov.cn/).

A total of 453 historical landslides with precise rainfall information, particular dates, and places were acquired by aggregating historical landslide data, removing landslides with no rainfall and missing rainfall data (refer to Fig. 3, Landslide).

The rainfall in the study area is mainly concentrated from May to October, and the differences in climatic conditions between the dry and wet seasons might result in various impacts of rainfall on landslide movement (Soralump et al., 2021). Therefore, in this study, according to the time of occurrence of historical landslides, landslides occurring from May to October are classified as rainy season landslides, while landslides occurring from November to April are classified as dry season landslides. According to the records, there were 412 rainy season landslides and 41 dry season landslides (Fig. 6). Among them, rainfall thresholds for rainy season landslides were calculated separately according to the sub-districts; whereas the number of dry season landslides is small and further subdivision is not conducive to the calculation of rainfall thresholds, so only rainfall thresholds for dry season landslides were calculated for the entire study area.




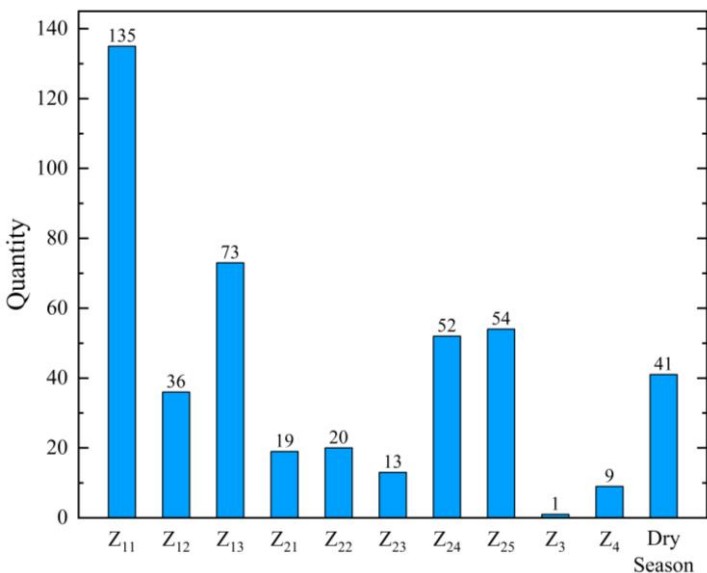

**Figure 6: Number of landslides in each sub-district in the rainy season and the whole region in the dry season.**

Figure 6 shows that the five zones $Z_{21}$, $Z_{22}$, $Z_{23}$, $Z_3$ and $Z_4$ have less catastrophe spots. To avoid insufficient data affecting rainfall threshold accuracy, this study merged some neighboring regions ($Z_{21}$ and $Z_{22}$ merged; $Z_{23}$, $Z_{24}$, and $Z_3$ merged; and

$Z_{25}$ and $Z_4$ merged) based on the geographic location of each region for rainfall threshold calculation.

## 4. Results

### 4.1 Rainfall Threshold Model Results

### 4.1.1 E-D Rainfall Threshold Model

Rainfall-triggered landslide is a random and small probability event, and if only the minimum threshold is used to warn of

geological hazards, it will produce many ineffective warnings (i.e., False Positive Error) (Sarkar et al., 2023). While decreasing the public's trust in disaster warning, it will result in a waste of resources for preventive and control activities, which is not favorable to the advancement of disaster prevention and mitigation. Therefore, most of the current studies on RTM use a variety of threshold curves with different landslide probabilities (Sheng et al., 2022), in order to improve the reasonableness and accuracy of rainfall warning. Generally, the landslide probability indicates the proportion of the number

of landslides triggered by rainfall exceeding a certain threshold among all occurring landslides (Yang et al., 2020).

In the calculation of OLS regression, the E and D scatters of historical landslide hazard locations in each area were first plotted into the E-D log-log coordinates system, and the 50% landslide probability rainfall threshold curve was derived by fitting using OLS regression. The fitted curves were then used to run OLS regression analysis on the historical landslide hazard points above and below the curves to get the 75% landslide probability rainfall threshold curve and the 25% landslide





probability rainfall threshold curve (Fig. 7). Finally, the log-log coordinates system straight lines were transformed to Cartesian coordinate system curves (Table 1).

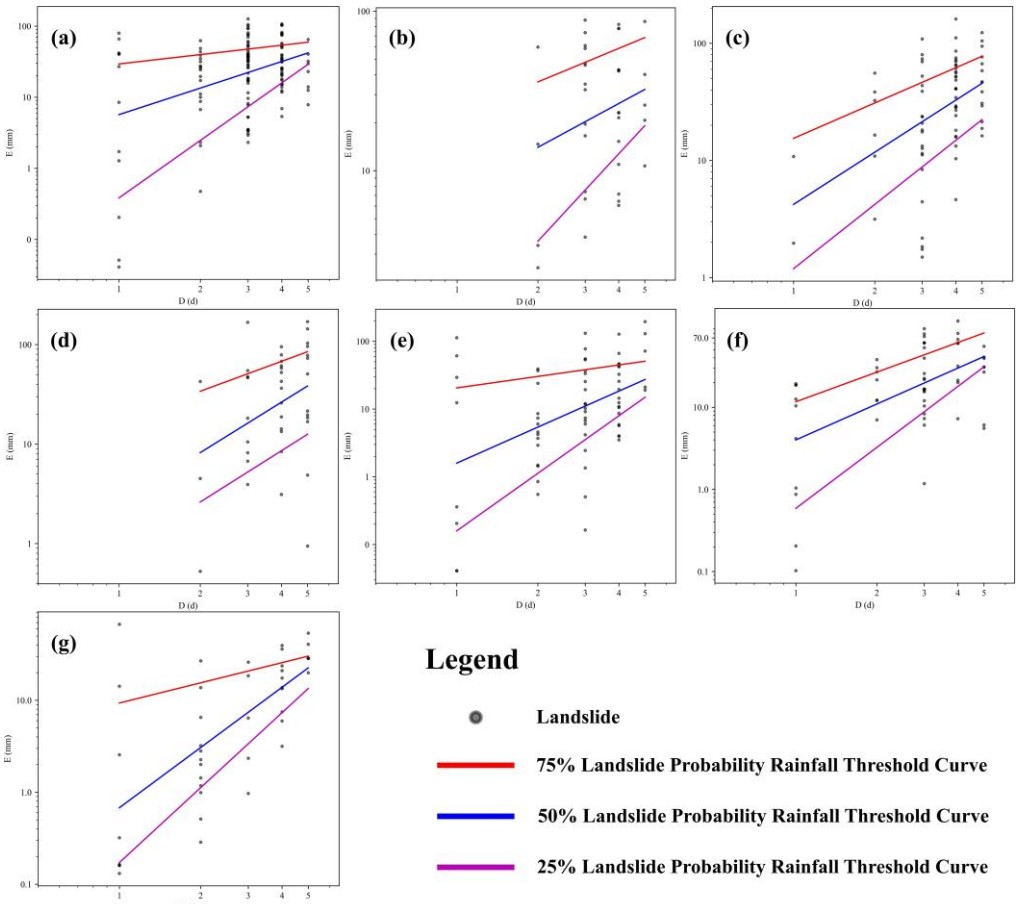

**Figure 7: Plot of E-D rainfall threshold model results in log-log coordinates system (OLS regression). In the figure, a is the $Z_{11}$ region, b is the $Z_{12}$ region, c is the $Z_{13}$ region, d is the $Z_{21}Z_{22}$ region, e is the $Z_{23}Z_{24}Z_3$ region, f is the $Z_{25}Z_4$ region, and g is the Dry Season**

**Table 1: E-D rainfall threshold equation (OLS regression).**

| Region | Landslide probability | Equations (Log-log coordinates system) | E-D equation |
|--------|----------------------|----------------------------------------|--------------|
| $Z_{11}$ | 75% | y=0.4383x+1.4679 | $E=29.3697 \times D^{0.4383}$ |
| | 50% | y=1.2420x+0.7552 | $E=5.6912 \times D^{1.2420}$ |
| | 25% | y=2.6894x-0.4164 | $E=0.3834 \times D^{2.6894}$ |
| $Z_{12}$ | 75% | y=0.6981x+1.3464 | $E=22.2024 \times D^{0.6981}$ |
| | 50% | y=0.9113x+0.8721 | $E=7.4490 \times D^{0.9113}$ |
| | 25% | y=1.8193x+0.0102 | $E=1.0238 \times D^{1.8193}$ |


| | | | |
|---|---|---|---|
| $Z_{13}$ | 75% | y=1.0019x+1.1887 | E=15.4419×D$^{1.0019}$ |
| | 50% | y=1.4792x+0.6246 | E=4.2131×D$^{1.4792}$ |
| | 25% | y=1.8201x+0.0759 | E=1.1910×D$^{1.8201}$ |
| $Z_{21}Z_{22}$ | 75% | y=0.9977x+1.2307 | E=17.0098×D$^{0.9977}$ |
| | 50% | y=1.6825x+0.4075 | E=2.5556×D$^{1.6825}$ |
| | 25% | y=1.7100x-0.0969 | E=0.8000×D$^{1.7100}$ |
| $Z_{23}Z_{24}Z_3$ | 75% | y=0.5633x+1.3125 | E=20.5353×D$^{0.5633}$ |
| | 50% | y=1.7673x+0.2014 | E=1.5900×D$^{1.7673}$ |
| | 25% | y=2.8230x-0.7986 | E=0.1590×D$^{2.8230}$ |
| $Z_{25}Z_4$ | 75% | y=1.1974x+1.0675 | E=11.6815×D$^{1.1974}$ |
| | 50% | y=1.4525x+0.6027 | E=4.0059×D$^{1.4525}$ |
| | 25% | y=2.4652x-0.2305 | E=0.5882×D$^{2.4652}$ |
| Dry Season | 75% | y=0.7295x+0.9706 | E=9.3454×D$^{0.7295}$ |
| | 50% | y=2.1754x-0.1679 | E=0.6794×D$^{2.1754}$ |
| | 25% | y=2.7079x-0.7646 | E=0.1719×D$^{2.7079}$ |

In the calculation of MLP regression, the rainfall thresholds corresponding to 50% landslide probability for each duration of rainfall (D) were first fitted separately. The MLP regression was then performed on the historical landslide data above and below the thresholds, respectively, to obtain the 75% landslide probability and 25% landslide probability rainfall thresholds

corresponding to each D. Due to the lack of historical landslide hazard data at a D of 1 in some regions (e.g., region $Z_{12}$) and the small amount of historical landslide hazard data at a D of 5 in some regions (e.g., region $Z_{11}$), these can lead to irrational results of the fitted rainfall thresholds. In this regard, this study used Gaussian regression (Kumar and Kavitha, 2021) and GM(1,1) grey prediction model (Chen and Huang, 2013) to correct the rainfall threshold results obtained from MLP regression. The corrected results are shown in Fig. 8 and Table 2.


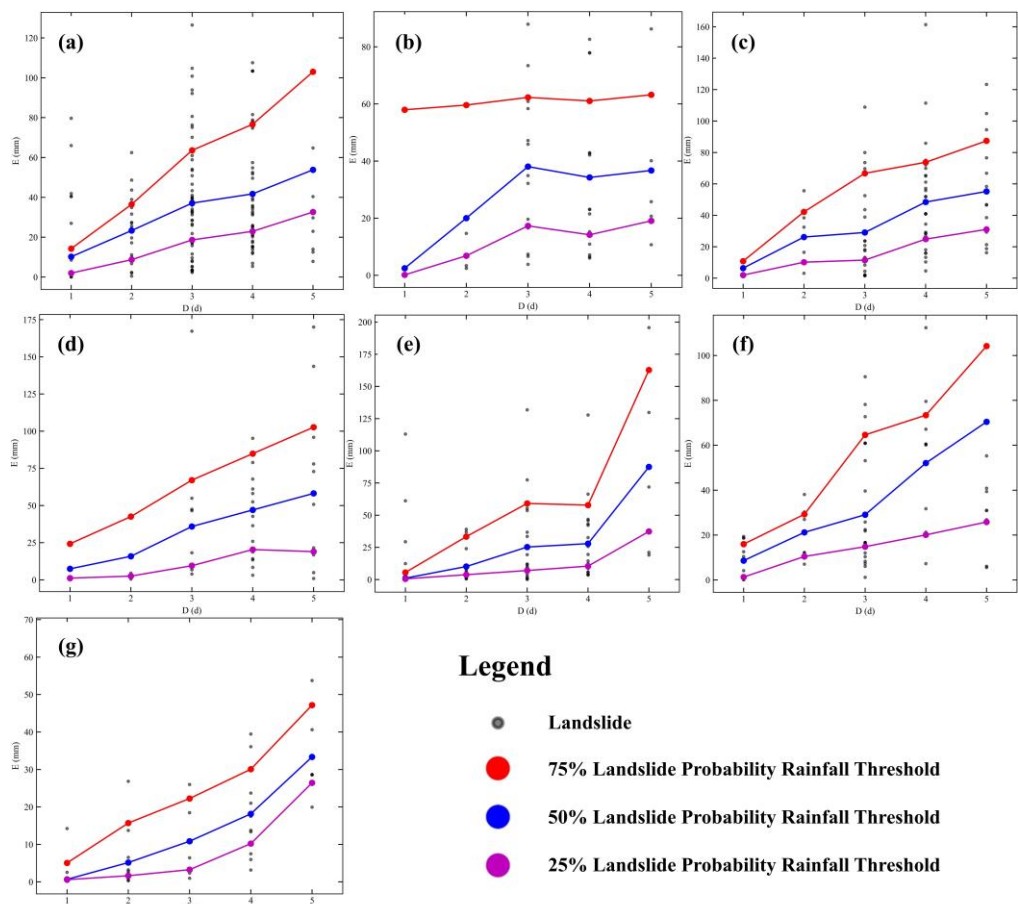


**Figure 8: Plot of E-D rainfall threshold model results (MLP regression). In the figure, a is the $Z_{11}$ region, b is the $Z_{12}$ region, c is the $Z_{13}$ region, d is the $Z_{21}Z_{22}$ region, e is the $Z_{23}Z_{24}Z_3$ region, f is the $Z_{25}Z_4$ region, and g is the Dry Season**

The red, blue, and purple points in Fig. 8 are the rainfall threshold points obtained from the fit for different landslide probabilities. The line segments are just for connecting the individual threshold points for viewing purposes and have no

**Table 2: E-D rainfall threshold (MLP regression).**

| Region | Duration of rainfall (D) | 75% threshold (mm) | 50% threshold (mm) | 25% threshold (mm) |
|---|---|---|---|---|
| $Z_{11}$ | 1 | 14.2305 | 10.1800 | 1.9625 |
|  | 2 | 36.4914 | 23.3267 | 8.7024 |
|  | 3 | 63.5907 | 37.0893 | 18.6210 |
|  | 4 | 76.6291 | 41.7210 | 22.9260 |
|  | 5 | 103.0000 | 53.8090 | 32.6260 |
| $Z_{12}$ | 1 | 57.9690 | 2.4749 | 0.1550 |
|  | 2 | 59.6126 | 20.0312 | 6.8458 |
|  | 3 | 62.3002 | 38.0666 | 17.3107 |





| | | | | |
|---|---|---|---|---|
| | 4 | 61.0451 | 34.2639 | 14.1966 |
| | 5 | 63.2107 | 36.7170 | 19.0748 |
| $Z_{13}$ | 1 | 10.8122 | 6.3897 | 1.9677 |
| | 2 | 42.1870 | 26.1761 | 10.1656 |
| | 3 | 66.7259 | 29.0723 | 11.5028 |
| | 4 | 73.7542 | 48.4590 | 24.8502 |
| | 5 | 87.3909 | 55.1944 | 31.0476 |
| $Z_{21}Z_{22}$ | 1 | 24.2575 | 7.4117 | 1.1585 |
| | 2 | 42.5658 | 15.8642 | 2.5160 |
| | 3 | 67.0825 | 35.8785 | 9.5152 |
| | 4 | 84.8807 | 47.0166 | 20.3769 |
| | 5 | 102.6789 | 58.1546 | 18.9942 |
| $Z_{23}Z_{24}Z_3$ | 1 | 5.5210 | 1.0893 | 0.5702 |
| | 2 | 33.3538 | 10.1252 | 3.7901 |
| | 3 | 59.1386 | 25.2715 | 7.0353 |
| | 4 | 57.8357 | 27.9044 | 10.4444 |
| | 5 | 162.7467 | 87.5204 | 37.3694 |
| $Z_{25}Z_4$ | 1 | 15.9482 | 8.6114 | 1.2742 |
| | 2 | 29.2418 | 21.1900 | 10.4545 |
| | 3 | 64.6284 | 29.0526 | 14.8209 |
| | 4 | 73.3920 | 52.0651 | 20.0756 |
| | 5 | 104.1990 | 70.4430 | 25.8100 |
| Dry Season | 1 | 5.0503 | 0.6647 | 0.5818 |
| | 2 | 15.7035 | 5.1495 | 1.6332 |
| | 3 | 22.2420 | 10.8428 | 3.2452 |
| | 4 | 30.0733 | 18.1523 | 10.2084 |
| | 5 | 47.1948 | 33.3588 | 26.4428 |

The threshold curves generated from OLS regression in the log-log coordinates system often exhibit an upward trend, as shown in Fig. 7, and the slopes of the rainfall threshold curves for 25%, 50%, and 75% landslide probability gradually decrease. From Fig. 8, the rainfall thresholds obtained from MLP regression for different landslide probabilities also show a generally increasing trend, but the relatively small amount of historical landslide data in some subregions results in relatively unreasonable rainfall thresholds (e.g., the rainfall threshold for the $Z_{23}Z_{24}Z_3$ region shows a large increase when D is 5).

### 4.1.2 E-D-R Rainfall Threshold Model

Based on the above E-D rainfall threshold model, the third dimension indicator R was introduced to construct the E-D-R rainfall threshold model. In this model, the value of R is taken equal to the rainfall threshold corresponding to when D is 1 in the E-D RTM. These three indicators visually form a closed "box" (Fig. 9), with "nested" relationships between the different landslide probability levels.





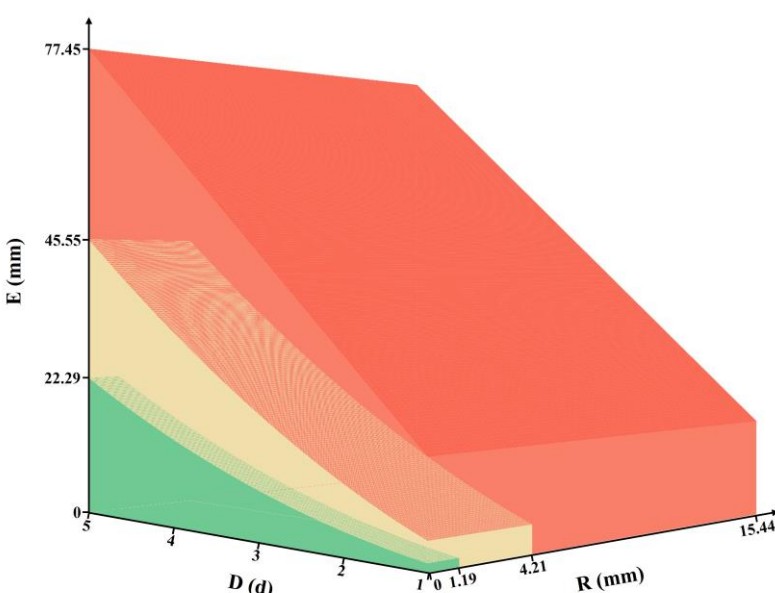

**Figure 9: Schematic of the E-D-R rainfall threshold model obtained from the OLS regression ($Z_{13}$).**

In Fig. 9, the green, yellow, and red boxes indicate rainfall thresholds of <25%, 25-50% and 50-75% landslide probability, respectively.

### 4.1.3 Model Accuracy Verification

The accuracy of the model was tested in this research utilizing 82 landslide hazards events that were not involved in the RTM calculations in 2019 and 2020. Figure 10 depicts the number of landslide hazards events in each region.

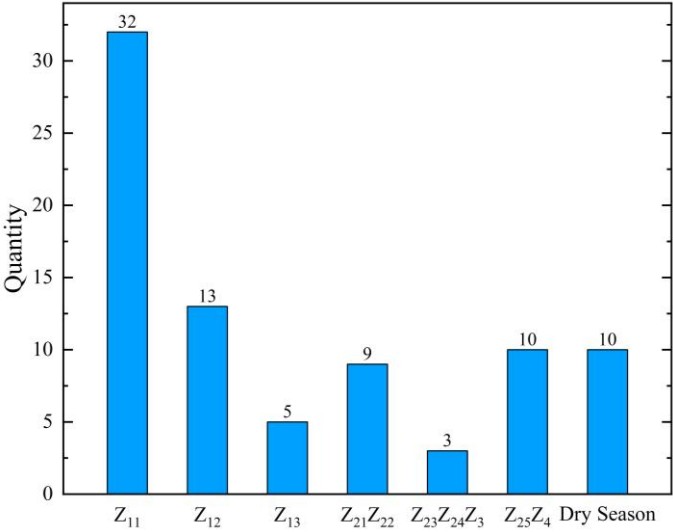

**Figure 10: The number of landslide hazard events in each region of the validation set.**





In the actual landslide control work, it is impossible to obtain the real rainfall on a certain day in the future, so it can only be replaced by the forecast rainfall. In order to make the validation data source of the rainfall threshold model more realistic, this study relies on the abundant rainfall forecasting stations in the study area (Fig. 11) and counts the forecast rainfall on the day of the occurrence of these 82 landslide hazards as well as the previous 5 days for the validation of the model.

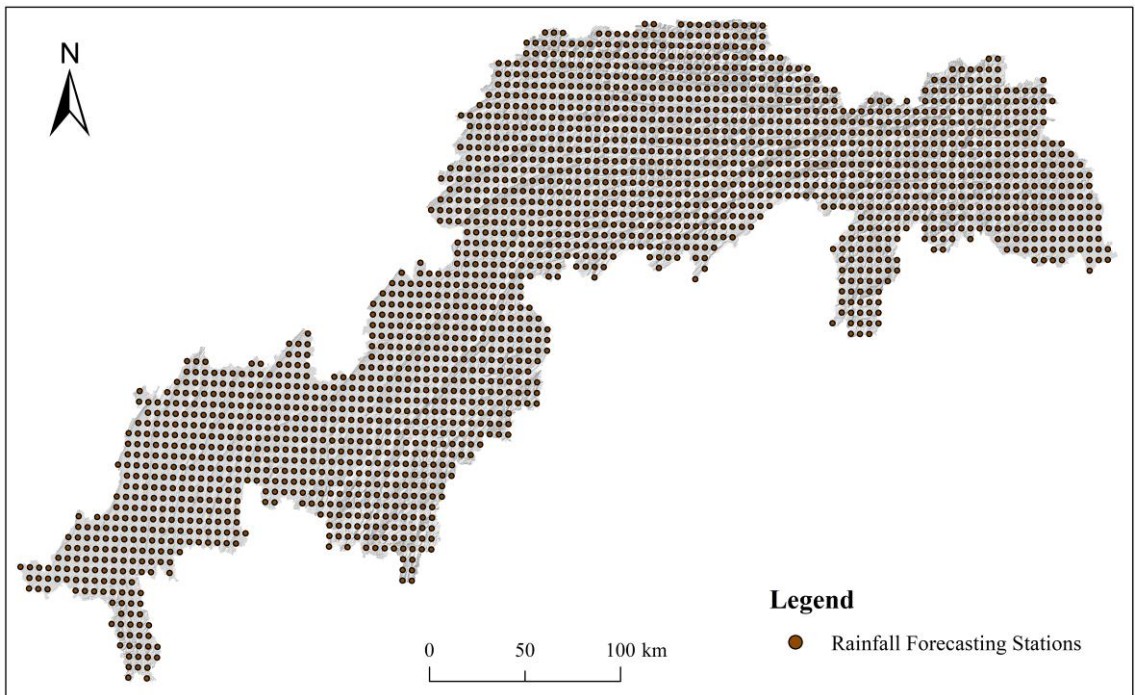

**Figure 11: Map of rainfall forecasting stations.**


The rainfall forecast stations in Fig. 11 are distributed at 0.05° intervals, and the forecast rainfall data were provided by the Wuhan Geological Survey Centre. The data are updated in real time according to meteorological changes, and the data used in the study are adopted from the latest update of the forecast data to ensure the accuracy of the data.

The research region was classified into four warning categories based on the rainfall threshold classification results: attention

(<25%), special attention (25-50%), warning (50%-75%), and severe warning (≥75%). Figure 12 displays the ultimate outcomes of the validation process for each region's four RTM categories. Furthermore, Table 3 displays the proportion of hazardous circumstances corresponding to the two warning levels of "severe warning" and "warning" in the E-D-R RTM validation results.




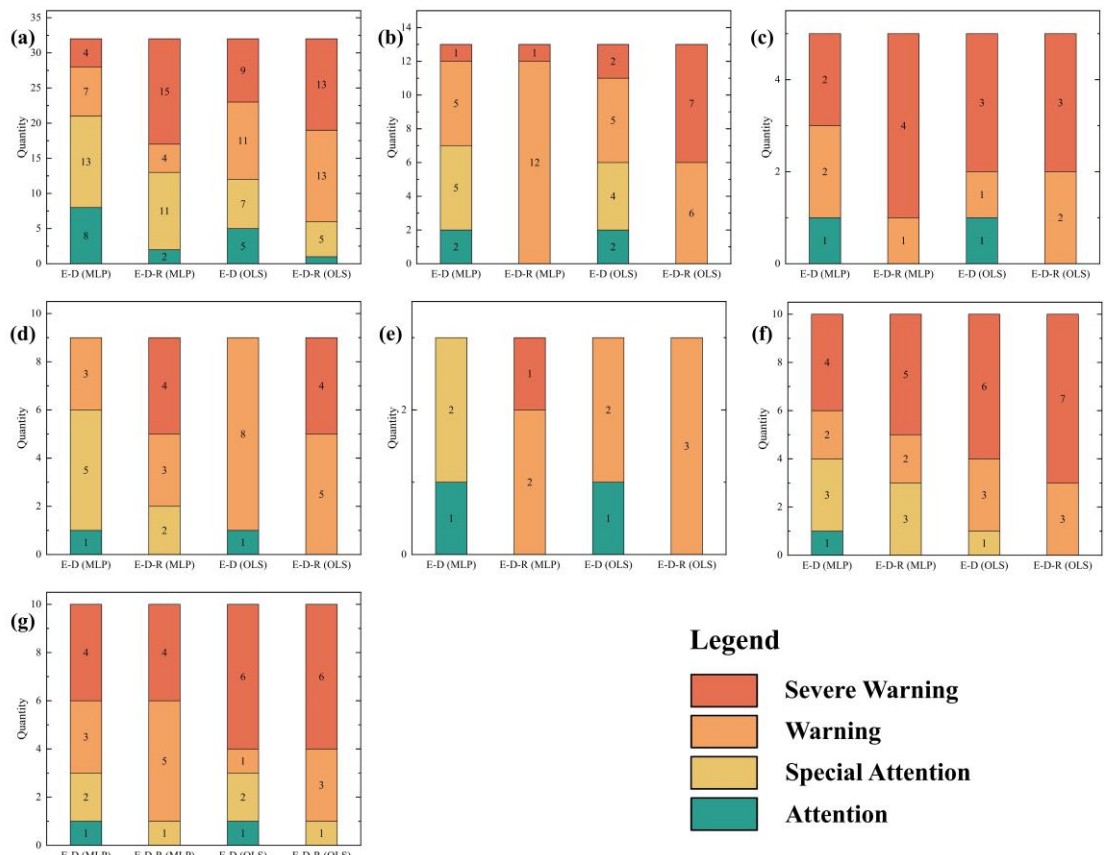

**Figure 12: The distribution of warning levels in the validation set for each partitioned region. In the figure, a is the $Z_{11}$ region, b is the $Z_{12}$ region, c is the $Z_{13}$ region, d is the $Z_{21}Z_{22}$ region, e is the $Z_{23}Z_{24}Z_3$ region, f is the $Z_{25}Z_4$ region, and g is the Dry Season**

**Table 3: Proportion of hazard events corresponding to the "Severe Warning" and "Warning" levels in the E-D-R RTM for each partitioned region.**

| Region | Regression approach | Level | Percentage (%) |
|---|---|---|---|
| $Z_{11}$ | MLP | Severe Warning | 46.88 |
| | | Warning | 12.50 |
| | OLS | Severe Warning | 40.63 |
| | | Warning | 40.63 |
| $Z_{12}$ | MLP | Severe Warning | 7.69 |
| | | Warning | 92.31 |
| | OLS | Severe Warning | 53.85 |
| | | Warning | 46.15 |
| $Z_{13}$ | MLP | Severe Warning | 80.00 |
| | | Warning | 20.00 |
| | OLS | Severe Warning | 60.00 |
| | | Warning | 40.00 |





| Region | Model | Warning Level | Value |
|---|---|---|---|
| $Z_{21}Z_{22}$ | MLP | Severe Warning | 44.44 |
| | | Warning | 33.33 |
| | OLS | Severe Warning | 44.44 |
| | | Warning | 55.56 |
| $Z_{23}Z_{24}Z_3$ | MLP | Severe Warning | 33.33 |
| | | Warning | 66.67 |
| | OLS | Severe Warning | 0.00 |
| | | Warning | 100.00 |
| $Z_{25}Z_4$ | MLP | Severe Warning | 50.00 |
| | | Warning | 20.00 |
| | OLS | Severe Warning | 70.00 |
| | | Warning | 30.00 |
| Dry Season | MLP | Severe Warning | 40.00 |
| | | Warning | 50.00 |
| | OLS | Severe Warning | 60.00 |
| | | Warning | 30.00 |

The following conclusions may be drawn from an analysis of the prediction accuracy of the four categories of RTM:

(1) The accuracies of the E-D-R RTM computed using MLP regression and OLS regression are much better than the comparable E-D RTM. The E-D-R RTM predict outputs no longer include the "Attention" warning level for all areas (Z11 excepted) when the R indicator was included in the third dimension. Furthermore, there has been a rise in the percentage of hazard incidents categorized as "Warning" and "Severe Warning" categories across all regions. Compared with the E-D model, the proportion of hazardous conditions in the "Warning" and "Severe Warning" warning levels of the E-D-R RTM

increases from 41.46% to 76.82%, and the result of OLS regression increases from 69.51% to 91.46%.

(2) The prediction accuracies of the E-D-R RTM for each region are slightly different between the MLP regression and the OLS regression, but in general, the total proportion of hazardous conditions at the warning levels of "Warning" and "Severe Warning" is similar.

(3) The optimal RTM for each region is shown in Table 4.

**Table 4: Optimal RTM for each partitioned region.**

| Region | Optimal rainfall threshold modelling (regression approach) |
|---|---|
| $Z_{11}$ | E-D-R (OLS) |
| $Z_{12}$ | E-D-R (OLS) |
| $Z_{13}$ | E-D-R (MLP) |
| $Z_{21}Z_{22}$ | E-D-R (OLS) |
| $Z_{23}Z_{24}Z_3$ | E-D-R (MLP) |





| | |
|---|---|
| $Z_{25}Z_4$ | E-D-R (OLS) |
| Dry Season | E-D-R (OLS) |

The optimal RTM for $Z_{13}$ and $Z_{23}Z_{24}Z_3$ regions are the E-D-R models obtained from the MLP regression, proving the feasibility of using neural networks (MLP) for RTM research.

## 4.2 Landslide Susceptibility Results

### 4.2.1 Landslide Inducing Factor Selection

Combined with the research results of previous scholars (Chen et al., 2021; Chen et al., 2020; Habumugisha et al., 2022; Li et al., 2022; Rohan et al., 2023) and the actual situation of the study area, a total of 11 landslide inducing factors, including elevation, Normalized Difference Vegetation Index (NDVI), Topographic Wetness Index (TWI), road density, stratigraphic lithology, tectonic density, river distance, slope, curvature, land cover, and slope structure, were selected in this study.

Table 5 shows the data sources for these 11 factors.

**Table 5: Source of data on landslide inducing factors.**

| Factor Category | Data Source | Inducing Factor |
|---|---|---|
| Topography and Geomorphology | Geological Map STRM DEM (30m) | Elevation |
| | | Slope |
| | | Curvature |
| | | Slope Structure |
| Geological Lithology | Geological Map | Stratigraphic Lithology |
| | | Tectonic Density |
| Hydrological Factor | National Basic Geographic Database STRM DEM (30m) | TWI |
| | | River Distance |
| Land Use | Landsat Remote Sensing Image (30m) | NDVI |
| | | Land Cover Type |
| Human Engineering Activities | OpenStreetMap | Road Density |

Among them, the slope structure considers the relationship between the slope aspect of the slope and the inclination of the rock formation (Niu et al., 2014), and different types of slope structures can lead to differences in landslide size and intensity. Based on different slope gradient ($\sigma$), slope direction ($\gamma$), and inclination ($\alpha$) and tendency ($\beta$) of the rock formation, the following eight types of slope structures are classified (Table 6).

**Table 6: Classification of slope structure types and percentage of each type in the study area.**

| Code | Relationship between $\alpha$, $\beta$, $\gamma$ and $\sigma$ | Area (%) |
|---|---|---|
| A | $\alpha \leq 5°$ | 1.720 |
| B | $\alpha > 5°$, $|\gamma-\beta| \in [0°, 30°)$ or $|\gamma-\beta| \in [330°, 360°)$, $\sigma > \alpha$ | 5.127 |





| C | $\alpha>5°$, $|\gamma-\beta|\in[0°, 30°)$ or $|\gamma-\beta|\in[330°, 360°)$, $\sigma=\alpha$ | 0.000 |
| D | $\alpha>5°$, $|\gamma-\beta|\in[0°, 30°)$ or $|\gamma-\beta|\in[330°, 360°)$, $\sigma<\alpha$ | 13.581 |
| E | $\alpha>5°$, $|\gamma-\beta|\in[30°, 60°)$ or $|\gamma-\beta|\in[300°, 330°)$ | 17.559 |
| F | $\alpha>5°$, $|\gamma-\beta|\in[60°, 120°)$ or $|\gamma-\beta|\in[240°, 300°)$ | 32.066 |
| G | $\alpha>5°$, $|\gamma-\beta|\in[120°, 150°)$ or $|\gamma-\beta|\in[210°, 240°)$ | 15.089 |
| H | $\alpha>5°$, $|\gamma-\beta|\in[150°, 210°)$ | 14.857 |

Stratigraphic lithology data was obtained by vectorizing and classifying geological maps (scale 1:200,000). Each lithology has a different pedogenic environment and will vary in composition and stability, which affects the occurrence of landslides (Cobos-Mora et al., 2023). In this paper, the study area is classified into four categories: carbonate, clastic, carbonate and clastic, as well as Igneous and metamorphic rocks. In addition, when the research area is large and most of the tectonics are

intertwined with each other, the distance from tectonics is no longer suitable as a correlation factor, and tectonic density should be used instead (Wang et al., 2014). Also, since the road data also show interlocking status, this paper uses tectonic density and road density as evaluation factors. When using ArcGIS to calculate the density, the search radius is kept as default, and the area unit is square kilometers.

To ensure the reasonableness of the selection of landslide inducing factors, this study used Pearson correlation analysis to

explore the degree of correlation among the selected inducing factors (Zhang et al., 2022) (Fig. 13). The value of correlation ranges from -1 to 1. The closer the value is to 1 or -1, the stronger the correlation between the two variables, and the closer the value is to 0, the weaker the correlation between the two variables (Cao et al., 2023).

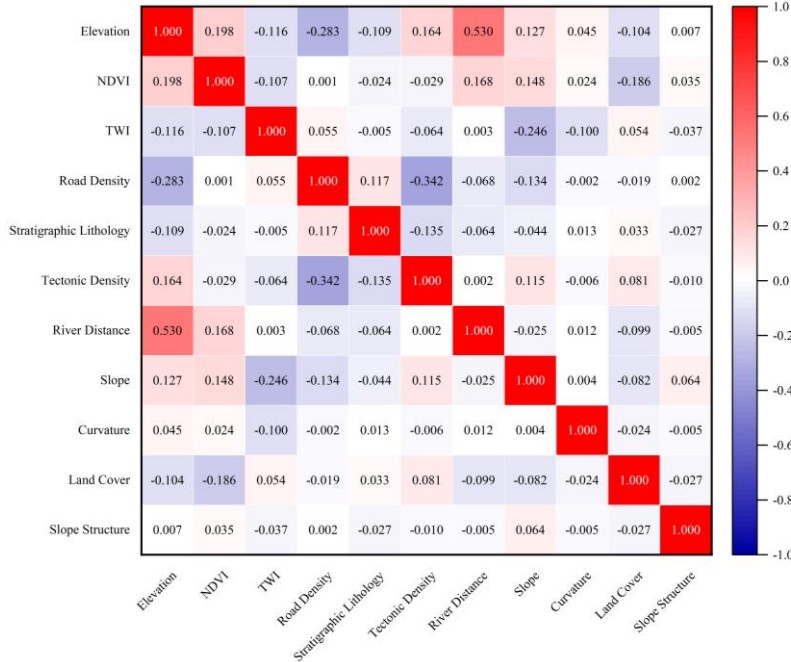

Figure 13: Pearson correlation results for inducing factors.



290 The correlation coefficients between the inducing factors are low, as shown in Fig. 13, with the exception of the somewhat higher correlation value between elevation and river distance (0.53). Given that elevation and river distance are two important factors for causing landslides (elevation is inherent in the assessment of LS (Wang et al., 2022b), which affects the distribution of submerged layers as well as the intensity of human activities; and the erosive effect of the river on the shoreline can damage the foot of the slope and soften the rock and soil mass (Selamat et al., 2022)), they are all retained in

295 this study. These 11 inducing factors were finally determined to be used in the TGRA's LS assessment research.

### 4.2.2 Grading of Landslide Susceptibility Factors

Combined with the actual situation of the study area and the results of previous studies, the class classification of each landslide predisposing factor and the result map of this study are shown in Table 7 and Fig. 14. The susceptibility evaluation was carried out in raster cells with a size of 30m × 30m. It's also worth noting that the historical landslide data utilized for LS

300 prediction includes all 6,888 recorded landslides, not just the 453 filtered for inclusion in the RTM calculations.

**Table 7: Classification of landslide inducing factors.**

| Predisposing Factor | Classification Criteria | Code |
|---|---|---|
| Elevation (m) | ≤300 | a |
| | (300,600] | |
| | (600,900] | |
| | (900,1200] | |
| | (1200,1500] | |
| | >1500 | |
| NDVI | [-1,0] | b |
| | (0,0.2] | |
| | (0.2,0.4] | |
| | (0.4,0.6] | |
| | (0.6,0.8] | |
| | (0.8,1] | |
| TWI | ≤6 | c |
| | (6,8] | |
| | (8,10] | |
| | (10,14] | |
| | >14 | |
| Road Density | [0,0.5] | d |
| | (0.5,1.2] | |
| | (1.2,2.5] | |
| | (2.5,5.0] | |
| | >5.0 | |





| | | |
|---|---|---|
| Stratigraphic Lithology | Carbonates | e |
| | Clastic rocks | |
| | Carbonates and clastic rocks | |
| | Igneous and metamorphic rocks | |
| Tectonic Density | [0,0.03] | f |
| | (0.03,0.12] | |
| | (0.12,0.24] | |
| | (0.24,0.38] | |
| | >0.38 | |
| River Distance (m) | ≤500 | g |
| | (500,1000] | |
| | (1000,1500] | |
| | >1500 | |
| Slope (°) | [0,10] | h |
| | (10,20] | |
| | (20,30] | |
| | (30,40] | |
| | (40,50] | |
| | >50 | |
| Curvature | ≤-3 | i |
| | (-3,-1] | |
| | (-1,0] | |
| | (0,1] | |
| | >1 | |
| Land Cover | Urban land | j |
| | Agricultural land | |
| | Forest land | |
| | Grassland | |
| | Water | |
| | Other Land | |
| Slope Structure | A | k |
| | B | |
| | D | |
| | E | |
| | F | |
| | G | |
| | H | |

**Figure 14: Landslide inducing factors grading results map.**

### 4.2.3 Landslide Susceptibility Evaluation Results

In this study, three models, CNN-3D, RF and SVM, were used to evaluate the LS of the study area, and the optimal LS result
was chosen for subsequent daily LHW. The relevant indicators obtained from the training of the three models are shown in
Table 8.

**Table 8: Results of the training of the susceptibility evaluation model.**

| Model | Model Evaluation Indicators | | | | |
|---|---|---|---|---|---|
| | AUC | Accuracy | Precision | Recall | F1_score |
| CNN-3D | 0.96 | 0.9003 | 0.8663 | 0.9295 | 0.8968 |
| RF | 0.82 | 0.7500 | 0.7656 | 0.7416 | 0.7534 |
| SVM | 0.83 | 0.7630 | 0.7625 | 0.7623 | 0.7624 |




Table 8 shows that the AUC values for CNN-3D, RF, and SVM models are 0.96, 0.82, and 0.83, respectively. The AUC
values indicate that all three models can better predict the probability of landslide occurrence in the study area, but the CNN-
3D model has a greater prediction accuracy than the RF and SVM models. In addition, for the other four metrics, the CNN-
3D model outperforms the RF and SVM models. As a consequence, in this study, the CNN-3D model's LS result was
divided into five classes using the natural breaks approach (Fig. 15) and was used for subsequent daily LHW.

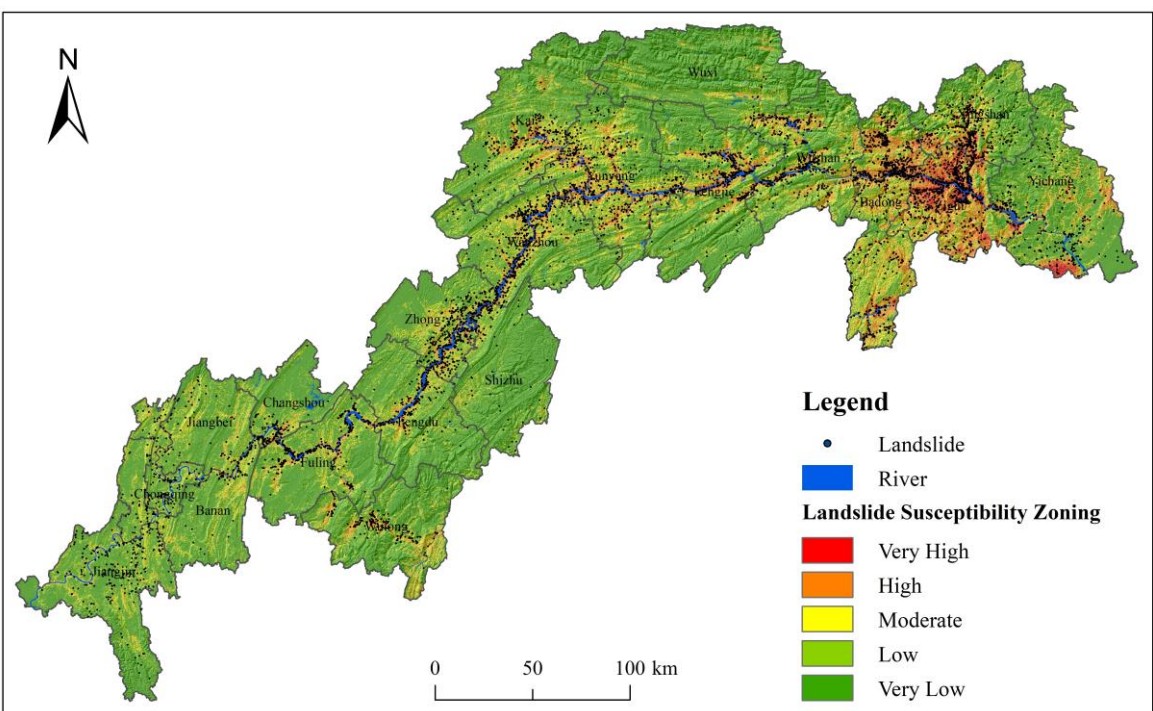

**Figure 15: CNN-3D model landslide susceptibility results.**

As a whole, the landslide disaster high susceptibility areas in the study area are mainly concentrated along the riverbanks and
in the central and eastern regions. In terms of district and county scopes, the landslide disaster high susceptibility areas are
mainly concentrated at Zigui, the northern part of Badong, the southern part of Xingshan, the central part of Fengjie, the
central part of Wanzhou, and the southeastern part of Zhongxian.

**4.3 Landslide Hazard Warning**

**4.3.1 Landslide Hazard Results for Each Rainfall Warning Level**

In this study, a superposition matrix (Table 9) was created to couple the daily RWL with the LS result to generate the daily
LHW result. Based on the superimposed matrix, four categories of landslide hazard levels will be obtained, where 1
indicates relatively stable zone, 2 indicates general prevention zone, 3 indicates secondary prevention zone, and 4 indicates
priority prevention zone.





**Table 9: Landslide susceptibility and rainfall warning level superposition matrix.**

| Susceptibility / Rainfall Threshold Level | Very Low | Low | Moderate | High | Very High |
|---|---|---|---|---|---|
| Caution | 1 | 1 | 1 | 1 | 2 |
| Special Caution | 1 | 1 | 1 | 2 | 3 |
| Warning | 1 | 1 | 2 | 3 | 4 |
| Severe Warning | 1 | 2 | 3 | 4 | 4 |

Based on the LS results shown in Fig. 15, combined with Table 9, the LHW results corresponding to each rainfall level were obtained (Fig. 16).

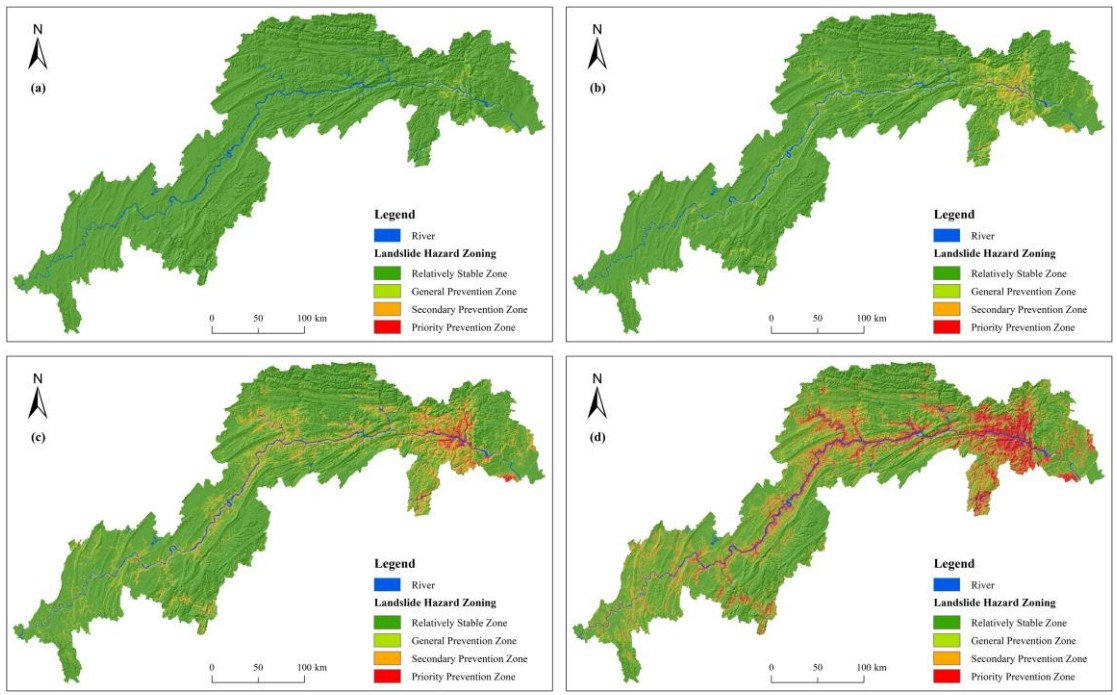

**Figure 16: Landslide hazard maps for each rainfall warning level. (a. attention level hazard; b. special attention level hazard; c. warning level hazard; d. severe warning level hazard).**

**4.3.2 Daily Landslide Hazard Warning**

In 2020, the Yangtze River experienced its worst basin-wide flood since 1998. on July 19, the "Yangtze River Flood No. 2 of 2020" was progressing through the TGRA to the middle and lower reaches of the Yangtze River, and the persistent rainfall induced many landslides. Therefore, in this study, 19 July 2020 was used as an example for LHW and validation. Based on the anticipated rainfall data at the time, E and D for the rainfall forecast stations from 14 July 2020 to 18 July 2020, and R for 19 July 2020, were calculated. Kriging interpolation was used to generate E (Fig. 17.a) and R (Fig. 17.b) for the whole research region. Since D is an integer ranging from 0 to 5, interpolation cannot be used to acquire D for the whole research




region; thus, this study uses the Thiessen polygon method and feature to raster method to obtain D for the entire study area

(Fig. 17.c).

The RWL for 19 July 2020 was calculated per sub-region (Fig. 17.d) using the optimum RTM for each sub-region obtained

above (Table 4).

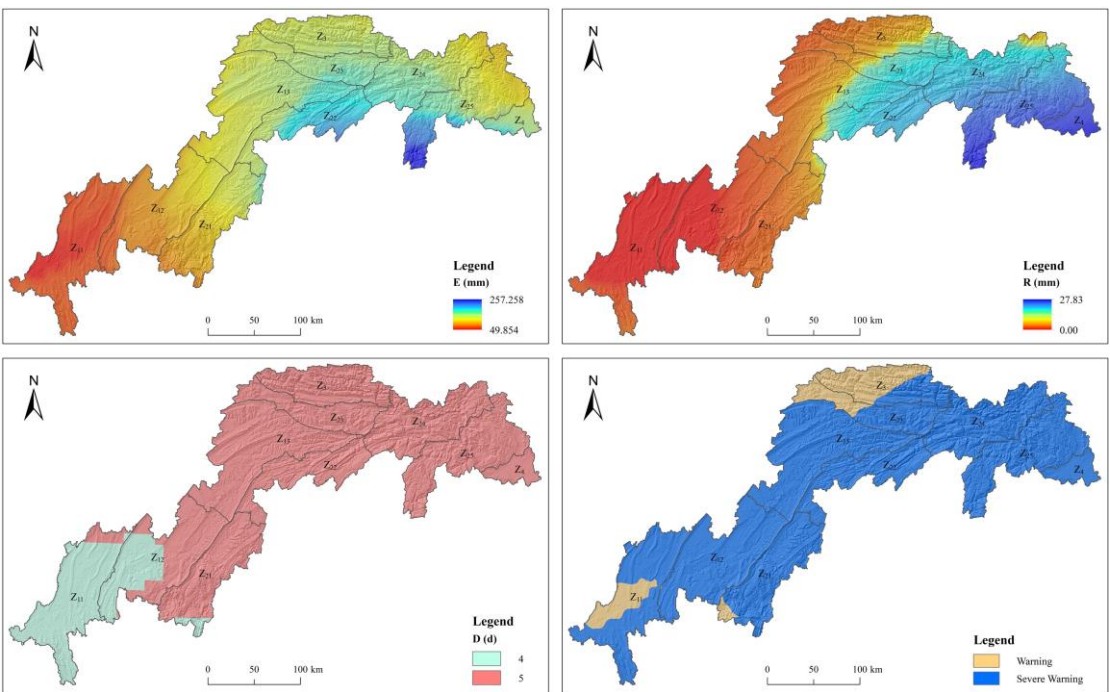

**Figure 17: Various rainfall parameters and rainfall warning levels for 19 July 2020.**

Based on the superposition matrix in Table 9, Fig. 17.d was superimposed on Figure 15 to obtain the LHW results for 19

July 2020 (Fig. 18).


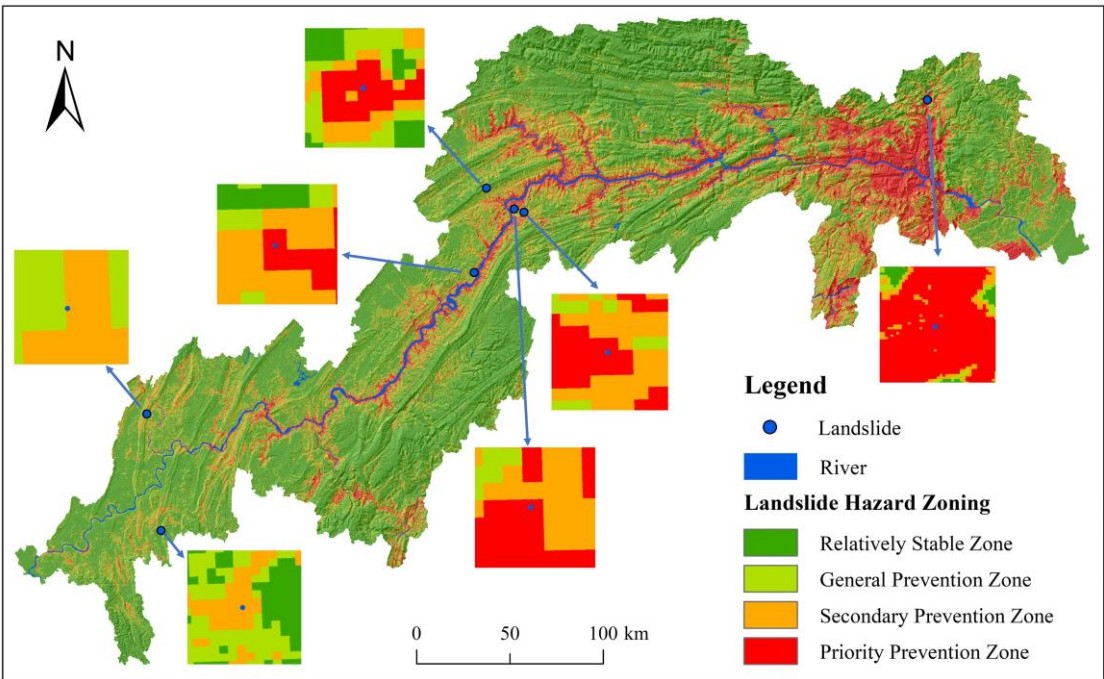

**Figure 18: Landslide hazard warning results for 19 July 2020.**

On July 19, 2020, there were seven landslide hazards, as shown in Fig. 18. Five of them fell in the priority prevention zone
and two in the secondary prevention zone, demonstrating the accuracy of both the LHW results and the rainfall threshold
model.

## 5. Discussion

### 5.1 Discussion of Rainfall Threshold Model

To investigate the best rainfall thresholds in the TGRA, two regression methods, OLS and MLP, and two RTM, E-D and E-
D-R, are used in this study. Regardless of the regression approach, the results reveal that the E-D-R model has greater
warning accuracy than the E-D model. In addition, the optimal RTM for two areas, $Z_{13}$ and $Z_{23}Z_{24}Z_3$, are the E-D-R models
obtained from the MLP regression, indicating the feasibility of using neural networks (MLP) for the study of RTM. However,
since the dataset of this study is not large (only 453 landslides) nor complex (only 3 variables), it may not be able to clearly
demonstrate the advantages of neural networks for rainfall threshold modeling. But we believe that this is a valuable attempt,
and more variables such as peak rainfall and rainfall intensity can be added in subsequent studies, and the application of
neural networks will certainly improve the accuracy of RWM.





To explore the reasons for the E-D-R model's higher warning accuracy, this study uses area $Z_{12}$ as an example, and shows some of the points where the RWL has been changed (i.e. landslides where the RWL has been increased) in the R-E plane view (Fig. 19), where the colors of the landslides indicate the different RWL, and the meaning is the same as in Fig. 12.

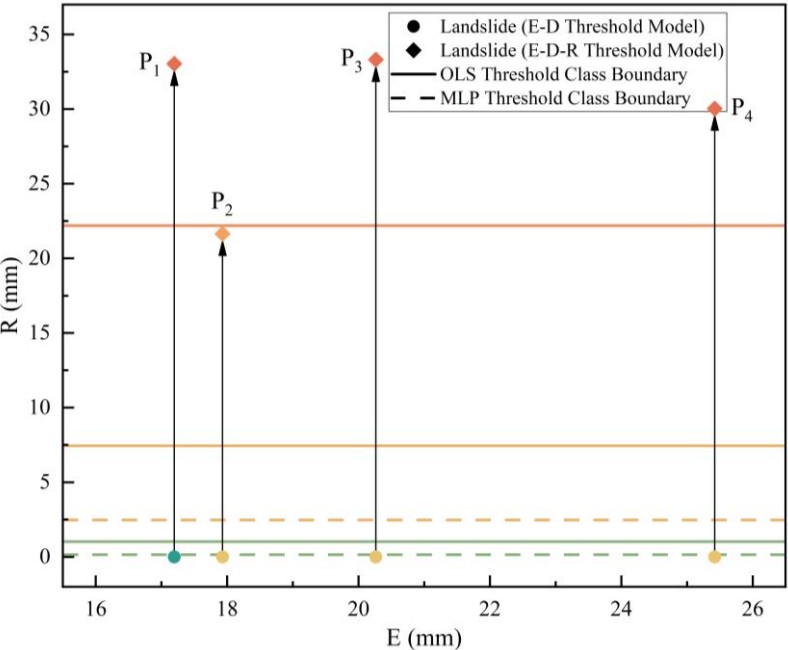


**Figure 19: Rainfall warning level transition process ($Z_{12}$ region).**

The chart shows that after the R indication was added, the RWL of the four landslides rose dramatically. The warning level of $P_1$ in the E-D model was only "Caution", and the warning levels of the remaining three landslides were only "Special Caution", whereas in the E-D-R model using OLS regression, the warning level of $P_2$ was raised to "Warning", and the
warning levels of the remaining three landslides were raised to "Severe Warning". Similarly, the alert levels of all four landslip points were raised to "Warning" in the E-D-R model using the MLP regression method. These landslides with RWL transition were the direct reason of the E-D-R model's improved accuracy in the $Z_{12}$ region.

Further exploration of the rainfall process of these four landslides before the landslide occurred (Fig. 20) reveals that these four landslides received less rainfall in the four days before the landslide, resulting in a lower E, but more rainfall on the day
of the landslide. The above characteristics make these four landslides have higher warning accuracy in the E-D-R RTM, indicating that the indicator R has some sensitivity in terms of landslides caused by heavy rain.


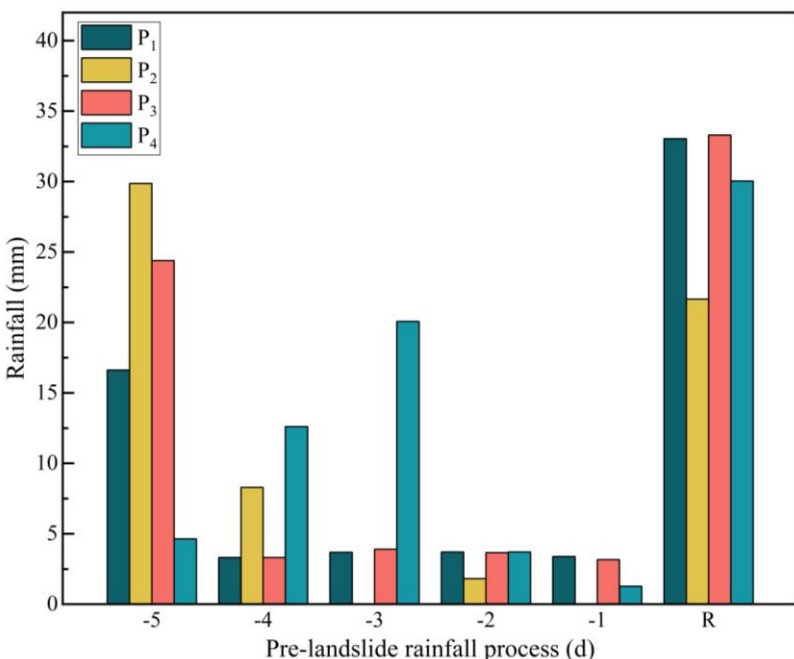

**Figure 20: Rainfall processes at rainfall warning level transition points.**

**5.2 Discussion of Daily Landslide Hazard Warning**

In this study, RF, SVM, and CNN-3D models were used to predict LS in the TGRA, and a comparison of the three models'
results showed that the CNN-3D model predicts LS with more accuracy in the study area. In addition, further analysis of the
CNN-3D model's LS results show that the very high LS zone is primarily distributed in areas with sparse vegetation, fragile
stratigraphic lithology, close to rivers, and active human engineering activities, which is similar with the results of Wang et
al (Wang et al., 2022a).

In terms of daily LHW, RWL are calculated using the optimal RTM for each sub-district based on forecast rainfall data from
rainfall stations. Subsequently, the daily LHW results were derived by utilizing a superposition matrix to combine the RWL
and LS results. On July 19, 2020, all seven landslide hazards are confirmed to be in the priority prevention and secondary
prevention zones. It can be observed that the LHW results obtained through the RTM have very high accuracy and are of
great significance in the prevention and control of landslide disasters. In addition, the process of transforming the LS results

into LHW results through the RWL and superposition matrix is essentially a correction process of the LS results. After the
correction, the areas that need to be focused on prevention and attention can be reduced to a certain extent, which saves the
cost of manpower and material resources in landslide prevention and control.





### 5.3 Practical Application of the Rainfall Threshold Model and Daily Landslide Hazard Warning

In the actual prevention and control of landslide hazards, it is inevitable to consider the factor of cost (Wang et al., 2023a).
To safeguard as many people's lives and property as possible within the limited cost range, it is necessary to narrow and refine the regions that must be prioritized while guaranteeing the accuracy of the LHW results.

The E-D-R RTM, while considering the advantages of the E-D RTM, increases the sensitivity to landslides induced by heavy rainfall on the same day, and has higher landslide warning accuracy. Meanwhile, the CNN-3D model fully considers the spatial information around each raster point, and its predicted LS results have higher prediction accuracy than those of the
RF and SVM models. Therefore, the E-D-R RTM and the CNN-3D model have a broad application space and development prospect in the warning and prevention of landslide disasters. The LHW results obtained by superposition of the results of the two models can ensure high accuracy and at the same time narrow down the areas that need to be focused on by virtue of the RWL results obtained by the RTM, so as to meet the requirements of landslide disaster prevention and control work.

In addition, although the E-D-R RTM as well as the CNN-3D model have high accuracy, there are certain uncertainties. For
the RTM: (1) The rainfall station can only accurately reflect the rainfall situation of the site, and there will be inaccuracies and uncertainties whether the rainfall data are extended to the whole study area by interpolation or Thiessen polygon method. (2) Historical landslide data play a decisive influence on the results of the rainfall threshold model. Either less historical landslide data or the existence of more extreme rainfall conditions will lead to uncertainty in the final RWL. (3) Although this study divided 10 regions as well as both dry and rainy seasons for the rainfall threshold study, the overall regional scope
is still large. There will be some uncertainty in the rainfall thresholds for different topography and geomorphology in the region. For the CNN-3D model, the selection of landslide-inducing factors, the size of the evaluation unit, the division ratio of the training set test set, and so on, will produce uncertainty in the results of LS.

Therefore, in the practical application of landslide prevention and control, it is necessary to combine the actual situation of the local area and select appropriate predisposing factors as well as evaluation units to ensure the accuracy of the LS results
(Zhang et al., 2023). Simultaneously, a historical landslide database can be constructed. When a new landslide occurs, the corresponding rainfall data will be summarized into the database and the rainfall threshold of the area will be recalculated for the subsequent RWL. The uncertainty of the RTM is expected to reduce as the quantity of historical landslide data grows, and the rainfall thresholds will continue to converge to the ultimate rainfall thresholds for the region. Furthermore, when the historical landslide data are sufficiently rich, the region may be split further to constantly improve the accuracy of the rainfall
warning level. Ultimately, the accuracy of LHW will be increased to give technical assistance for subsequent assessment of vulnerability as well as disaster preventive and mitigation efforts.


## 6. Conclusion

Landslide disaster warning is an essential tool in the prevention and management of landslides. To improve the accuracy of landslide warning, this paper first chose two regression methods, MLP and OLS, and two RTM, E-D and E-D-R, and divided

the TGRA into two dry and rainy seasons, as well as several sub-districts based on topography and rainfall, to explore the optimal RTM for the study area and obtain the daily RWL. Subsequently, 11 inducing factors were selected to investigate the LS in the study area utilizing three models: RF, SVM, and CNN-3D. Finally, using a superposition matrix, the RWL was overlaid on the LS results to achieve daily LHW in the TGRA.

In terms of rainfall threshold models, the study's results suggest that the E-D-R RTM has superior sensitivity in terms of

landslides induced by heavy rainfall, therefore the rainfall warning accuracy produced by either regression method is higher than that of the E-D model. In addition, for each sub-district, the optimal RTM for the four zones $Z_{11}$, $Z_{12}$, $Z_{21}Z_{22}$, $Z_{25}Z_4$, and Dry Season is the E-D-R RTM calculated by OLS regression; whereas the optimal RTM for the two zones $Z_{13}$ and $Z_{23}Z_{24}Z_3$ is the E-D-R RTM obtained by MLP regression. In terms of LS, the CNN-3D model's AUC and Accuracy achieved 0.96 and 0.9003, respectively, and its prediction accuracy outperformed the RF and SVM models.

The daily LHW is calculated by combining the daily RWL and the landslide susceptibility results. Data from the 19 July 2020 hazard event were utilized to verify the LHW results in this research. Of the seven landslide hazards on that date, five fell in the priority prevention zone and two in the secondary prevention zone, proving the accuracy of the LHW results and the RTM.

The RTM was utilized to obtain the daily RTL, and then overlaid with the LS results to obtain the daily LHW, which may be

used as guidance and reference for local landslide disaster prevention and control operations. In addition, the introduction of MLP to regression analysis of rainfall threshold in this study also further enriches the calculation method of RTM, which is of some significance for promotion.

**Code and data availability**

The data and code can be accessed at https://doi.org/10.5281/zenodo.11311851 (Peng, 2024).

**Author contributions**

**Bo Peng**: Writing - original draft, Writing - review & editing, Data curation, Formal analysis, Validation.

**Xueling Wu**: Writing - review & editing, Funding acquisition, Conceptualization, Methodology.



**Competing interests**

The authors declare that they have no conflict of interest.

**Disclaimer**

Publisher's note: Copernicus Publications remains neutral with regard to jurisdictional claims made in the text, published
maps, institutional affiliations, or any other geographical representation in this paper. While Copernicus Publications makes
every effort to include appropriate place names, the final responsibility lies with the authors.

**Acknowledgements**

This study was supported by the National Natural Science Foundation of China (42071429).

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
