# Peer review of "Optimizing Rainfall-Triggered Landslide Thresholds to Warning Daily Landslide Hazard in Three Gorges Reservoir Area"

_Natural Hazards and Earth System Sciences, 2024_

## Author Comment (AC1)

**Responses to Reviewer:**

*General comments: This paper proposes a method for calculating rainfall thresholds for rainfall-induced landslides using the Multilayer Perceptron regression method and the feasibility of this method has been verified. In addition, the authors use a large amount of data and various data-driven modeling techniques. The research results have practical significance for the early warning and prevention of rainfall-induced landslides. It is recommended that the paper be published after revisions, addressing the following comments:*

**Response**: We thank you for your recommendation and valuable comments that have ultimately improved this manuscript. We greatly appreciate you for the very extensive and thoughtful review of the manuscript. According to your comments, we have made point by point corrections which we hope meet with your approval.

**Point by point responses to the nine comments**:

*1. Comment: The originality of the study is not prominently highlighted in the abstract. It is recommended to enhance this aspect.*

**Response**: Thank you for pointing this out. We understand the concern regarding the lack of emphasis on the originality of the study in the abstract. Based on your suggestion, we have revised the abstract to highlight the originality of our research. The revised abstract can be found in lines 6-18 (in red font).

**Abstract.** Rainfall is intrinsically connected to the incidence of landslide catastrophes.

Exploring the ideal rainfall threshold model (RTM) for an area to determine the rainfall warning level (RWL) for daily landslide hazard warning (LHW) is critical for precise prevention and management of local landslides. In this paper, we propose a novel approach using multilayer perceptron (MLP) regression to calculate rainfall thresholds for 453 rainfall-induced landslides. This is the first study to integrate MLP and ordinary least squares (OLS) methods to determine the optimal RTM for distinct subregions, which were divided based on topography and climate conditions. Additionally, we introduce an innovative application of a three-dimensional convolutional neural network (3D-CNN) model to predict landslide susceptibility (LS) with higher accuracy. Finally, we develop a comprehensive methodology to overlay daily RWL with LS predictions using a superposition matrix, providing daily LHW results for the study area. The study's findings are: (1) The optimal RTMs and calculation methods vary across different subregions, highlighting the necessity of tailored approaches. (2) The 3D-CNN model significantly enhances LS prediction accuracy. (3) The daily LHW was validated using anticipated rainfall data for July 19, 2020, demonstrating the reliability of the LHW results and RTM. This study offers a significant advancement in the precise prediction and management of landslide hazards through innovative modeling techniques.

***2. Comment****: The use of the Multilayer Perceptron (MLP) for analyzing rainfall thresholds is a commendable innovation. However, since this method has been widely used in other fields, it would suffice to mention it with appropriate references. The MLP framework in Figure 1 is relatively simple and takes up significant space; consider removing it.*

**Response**: We completely agree with your suggestion. The Multilayer Perceptron (MLP) is a significant model in machine learning and has been widely applied in various research fields by scholars both domestically and internationally. Given that the description of the MLP framework occupies considerable space, we have removed the related basic description and the framework diagram from the revised manuscript.

***3. Comment****: Given the length of the article and the complexity of the methods and procedures involved, it is suggested that the authors create a flowchart to further elucidate the methodological steps.*

**Response**: Thank you for bringing this to our attention. We understand your concern regarding the complexity of the content and the lack of a flowchart. To more clearly illustrate the methods and steps involved in the article, we have created a flowchart and included it in the revised manuscript (lines 71-73, in red font).

The flowchart for the study is shown in Fig. 1.

[Figure]

**Figure 1. Flowchart of this study.**

***4. Comment****: The paper mentions dividing the study area based on topography and climate, followed by partial merging based on the number of historical disasters. It is suggested to include the final regional division results in Figure 4*

*to avoid any ambiguity.*

**Response**: Your feedback is greatly appreciated and has been very helpful for improving our work. We agree with your suggestion; the absence of the final merged regional division in Figure 4 may cause confusion for readers. Therefore, we have added black outlines and labels to show the final regional divisions in the figure. Additionally, following your seventh suggestion, we have combined Figure 6 with Figure 4. The revised content can be found in lines 147-149 (in red font).

[Figure]

**Figure 4: Zoning map of the study area. (a) Schematic diagram of the sub-region merger; (b) The number of historical landslide hazard sites in each sub-region.**

**5. Comment**: *In Table 6, the categories of slope structures are represented by A-H, which is unclear and it is recommended to change them to professional terms.*

**Response**: Thank you for pointing this out. As you mentioned, using A-H to

represent different slope structures may be misleading to readers. We have

revised the categories of slope structures in Tables 6 and 7 using professional

terminology. The revised content can be found in lines 271 and 295 (in red

font).

**Table 6: Classification of slope structure types and percentage of each type in the study area.**

| Class | Relationship between α, β, γ and σ | Area (%) |
|---|---|---|
| Nearly horizontal slope | α≤5° | 1.720 |
| Over-dip slope | α>5°, \|γ-β\|∈[0°, 30°) or \|γ-β\|∈[330°, 360°), σ>α | 5.127 |
| Flat-dip slope | α>5°, \|γ-β\|∈[0°, 30°) or \|γ-β\|∈[330°, 360°), σ=α | 0.000 |
| Under-dip slope | α>5°, \|γ-β\|∈[0°, 30°) or \|γ-β\|∈[330°, 360°), σ<α | 13.581 |
| Dip-oblique slope | α>5°, \|γ-β\|∈[30°, 60°) or \|γ-β\|∈[300°, 330°) | 17.559 |
| Transverse slope | α>5°, \|γ-β\|∈60°, 120°) or \|γ-β\|∈[240°, 300°) | 32.066 |
| Anticlinal-oblique slope | α>5°, \|γ-β\|∈[120°, 150°) or \|γ-β\|∈[210°, 240°) | 15.089 |
| Anticlinal slope | α>5°, \|γ-β\|∈[150°, 210°) | 14.857 |

**6. Comment**: *In Table 7, the units of some landslide susceptibility factors are given,*

*but the units for factors such as road density are missing.*

**Response**: Thank you for bringing this to our attention. We apologize for the

omission of units for landslide susceptibility factors such as road density. In

the revised manuscript, we have included the missing units (line 295, in red

font).

**Table 7: Classification of landslide inducing factors (only the revised part is shown).**

| Predisposing Factor | Classification Criteria | Code |
|---|---|---|
| Road Density (km/km$^2$) | [0,0.5] | d |
| | (0.5,1.2] | |
| | (1.2,2.5] | |
| | (2.5,5.0] | |
| | >5.0 | |
| Tectonic Density (km/km$^2$) | [0,0.03] | f |
| | (0.03,0.12] | |
| | (0.12,0.24] | |

| | | |
|---|---|---|
| | (0.24,0.38] | |
| | >0.38 | |
| | ≤-3 | |
| | (-3,-1] | |
| Curvature (m$^{-1}$) | (-1,0] | i |
| | (0,1] | |
| | >1 | |
| | Nearly horizontal slope | |
| | Over-dip slope | |
| | Under-dip slope | |
| Slope Structure | Dip-oblique slope | k |
| | Transverse slope | |
| | Anticlinal-oblique slope | |
| | Anticlinal slope | |

*7. **Comment***: *There are many images in the article. Consider combining some of them for display.*

**Response**: Thank you for your suggestion. The article contains a substantial amount of content, resulting in a large number of images and tables, which has made the manuscript quite lengthy. Additionally, some figures provide limited information. Therefore, we have combined the Thiessen polygon results from Figure 5 into Figure 3, and merged the historical disaster point bar chart from Figure 6 with Figure 4. We also removed Figure 11, which depicted the rainfall forecast stations, as it did not present useful information. The revised content can be found in lines 136-137 and 147-149 (in red font).

[Figure]

**Figure 3: Geographic location of the study area and Thiessen polygon results of rainfall stations.**

[Figure]

**Figure 4: Zoning map of the study area. (a) Schematic diagram of the sub-region merger; (b) The number of historical landslide hazard sites in each sub-region.**

*8. Comment: Some descriptions of the figures, such as the explanation of different*

*colors in Figure 9 in lines 223-224, should be moved from the main text to the*

*figure captions.*

**Response**: Thank you for pointing this out. We apologize for placing some explanatory notes that should be in the figure or table captions within the main text, which has made the text unnecessarily lengthy. We have reviewed and revised the captions for all figures and tables accordingly. The affected figures and tables include Figure 6 (formerly Figure 8), Figure 7 (formerly Figure 9), Table 9, Figure 14 (formerly Figure 17), and Figure 16 (formerly Figure 19). The revised content can be found in lines 204-208, 220-223, 318-319, 329-332, and 353-356 (in red font).

[Figure]

**Figure 6:** Plot of E-D rainfall threshold model results (MLP regression). In the figure, a is the $Z_{11}$ region, b is the $Z_{12}$ region, c is the $Z_{13}$ region, d is the $Z_{21}Z_{22}$ region, e is the $Z_{23}Z_{24}Z_3$ region, f is the $Z_{25}Z_4$ region, and g is

**the Dry Season. The red, blue and purple points in the figure are the rainfall threshold points obtained by fitting different landslide probabilities. The line segments are only used to connect the threshold points for viewing and have no practical significance.**

[Figure]

**Figure 7: Schematic diagram of the E-D-R rainfall threshold model in Z13 region obtained by OLS regression. The green, yellow and red boxes in the figure represent the landslide probability when the rainfall threshold is <25%, 25-50% and 50-75%, respectively.**

**Table 9: Landslide susceptibility and rainfall warning level superposition matrix. In the table, 1 indicates relatively stable zone, 2 indicates general prevention zone, 3 indicates secondary prevention zone, and 4 indicates priority prevention zone.**

| Rainfall Threshold Level \ Susceptibility | Very Low | Low | Moderate | High | Very High |
|---|---|---|---|---|---|
| Caution | 1 | 1 | 1 | 1 | 2 |
| Special Caution | 1 | 1 | 1 | 2 | 3 |
| Warning | 1 | 1 | 2 | 3 | 4 |
| Severe Warning | 1 | 2 | 3 | 4 | 4 |

[Figure]

**Figure 14: Various rainfall parameters and rainfall warning levels on July 19, 2020. (a) Effective rainfall obtained by Kriging interpolation; (b) Rainfall for the day obtained by Kriging interpolation; (c) Duration of rainfall obtained by Thiessen polygons; (d) Rainfall warning level calculated by the optimal rainfall threshold model.**

[Figure]

**Figure 16: Rainfall warning level transition process ($Z_{12}$ region). Green is the dividing line between Special Attention and Attention levels; yellow is the dividing line between Warning and Special Attention levels; orange is the dividing line between Severe Warning and Warning levels.**

**9. Comment**: *The clarity of Figure 14 is insufficient. It is recommended to change*

*the layout from three columns to two columns.*

**Response**: Thank you for bringing this to our attention. Following your suggestion, we have changed the layout of the figure from three columns to two columns to improve clarity. The revised content can be found in lines 296-297 (in red font).

[Figure]

**Figure 11: Landslide inducing factors grading results map.**

*Special thanks to you for your insightful and valuable comments in detail.*

---

## Author Comment (AC2)

**Responses to Reviewer:**

*General comments: This paper proposes a method for calculating rainfall thresholds for rainfall-induced landslides using the Multilayer Perceptron regression method and the feasibility of this method has been verified. In addition, the authors use a large amount of data and various data-driven modeling techniques. The research results have practical significance for the early warning and prevention of rainfall-induced landslides. It is recommended that the paper be published after revisions, addressing the following comments:*

**Response**: We thank you for your recommendation and valuable comments that have ultimately improved this manuscript. We greatly appreciate you for the very extensive and thoughtful review of the manuscript. According to your comments, we have made point by point corrections which we hope meet with your approval.

**Point by point responses to the nine comments**:

*1. Comment: The originality of the study is not prominently highlighted in the abstract. It is recommended to enhance this aspect.*

**Response**: Thank you for pointing this out. We recognize the issue of not sufficiently highlighting the originality of our study in the abstract. Based on your suggestion, we have revised the abstract to better emphasize the originality of our research. The revised abstract can be found in lines 6-20 (in red font).

**Abstract.** Rainfall is intrinsically linked to the occurrence of landslide catastrophes.

Identifying the most suitable rainfall threshold model for an area is crucial for establishing effective daily landslide hazard warnings, which are essential for the precise prevention and management of local landslides. This study introduces a novel approach that utilizes multilayer perceptron (MLP) regression to calculate rainfall thresholds for 453 rainfall-induced landslides. This research represents the first attempt to integrate MLP and ordinary least squares methods for determining the optimal rainfall threshold model tailored to distinct subregions, categorized by topographical and climatic conditions. Additionally, an innovative application of a three-dimensional convolutional neural network (CNN-3D) model is introduced to enhance the accuracy of landslide susceptibility predictions. Finally, a comprehensive methodology is developed to integrate daily rainfall warning levels with landslide susceptibility predictions using a superposition matrix, thus offering daily landslide hazard warning results for the study area. The key findings of this study are as follows: (1) The optimal rainfall threshold models and calculation methods vary across different subregions, underscoring the necessity for tailored approaches. (2) The CNN-3D model substantially improves the accuracy of landslide susceptibility predictions. (3) The daily landslide hazard warnings were validated using anticipated rainfall data from July 19, 2020, thereby demonstrating the reliability of both the landslide hazard warning results and the rainfall threshold model. This study presents a substantial advancement in the precise prediction and management of landslide hazards by employing innovative modeling techniques.

**2. Comment**: *The use of the Multilayer Perceptron (MLP) for analyzing rainfall thresholds is a commendable innovation. However, since this method has been widely used in other fields, it would suffice to mention it with appropriate references. The MLP framework in Figure 1 is relatively simple and takes up significant space; consider removing it.*

> **Response**: We fully agree with your suggestion. MLP is an important model in machine learning and has been widely applied in various research fields by scholars worldwide. Given that the description of the MLP framework in this paper occupied considerable space, we have removed the related basic description and framework diagram in the revised manuscript.

**3. Comment**: *Given the length of the article and the complexity of the methods and procedures involved, it is suggested that the authors create a flowchart to further elucidate the methodological steps.*

> **Response**: Thank you for bringing this to our attention. We understand your concern regarding the complexity of the content and the lack of a flowchart. To more clearly illustrate the methods and steps involved in the study, we have created a flowchart and included it in the revised manuscript (lines 72-74, in red font).

The flowchart for the study is shown in Fig. 1.

[Figure]

**Figure 1. Flowchart of this study.**

*4. Comment: The paper mentions dividing the study area based on topography and climate, followed by partial merging based on the number of historical disasters. It is suggested to include the final regional division results in Figure 4 to avoid any ambiguity.*

**Response**: Your feedback has been extremely helpful in improving our work. We fully agree with your suggestion that the final regional division results were not displayed in Figure 3 (previously Figure 4), which could cause confusion for readers. Therefore, we have added black outlines and labels to the figure to show the final regional divisions. Additionally, following your seventh suggestion, we have also merged the original Figure 6 with it. The revised content can be found in lines 152-154 (in red font).

[Figure]

**Figure 3: Zoning map of the study area. (a) Schematic diagram of the sub-region merger; (b) Number of historical landslide hazard sites in each sub-region.**

*5. **Comment***: *In Table 6, the categories of slope structures are represented by A-H, which is unclear and it is recommended to change them to professional terms.*

**Response**: Thank you for pointing this out. As you mentioned, using A-H to represent different slope structures could indeed be misleading for readers. We have revised the categories of slope structures in Tables 6 and 7 to use professional terminology. The revised content can be found in lines 262 and 287 (in red font).

**Table 6: Classification of slope structure types and their respective percentages within the study area.**

| Class | Relationship between α, β, γ and σ | Area (%) |
|---|---|---|
| Nearly horizontal slope | α≤5° | 1.720 |
| Over-dip slope | α>5°, \|γ-β\|∈[0°, 30°) or \|γ-β\|∈[330°, 360°), σ>α | 5.127 |
| Flat-dip slope | α>5°, \|γ-β\|∈[0°, 30°) or \|γ-β\|∈[330°, 360°), σ=α | 0.000 |
| Under-dip slope | α>5°, \|γ-β\|∈[0°, 30°) or \|γ-β\|∈[330°, 360°), σ<α | 13.581 |
| Dip-oblique slope | α>5°, \|γ-β\|∈[30°, 60°) or \|γ-β\|∈[300°, 330°) | 17.559 |
| Transverse slope | α>5°, \|γ-β\|∈60°, 120°) or \|γ-β\|∈[240°, 300°) | 32.066 |
| Anticlinal-oblique slope | α>5°, \|γ-β\|∈[120°, 150°) or \|γ-β\|∈[210°, 240°) | 15.089 |
| Anticlinal slope | α>5°, \|γ-β\|∈[150°, 210°) | 14.857 |

**Table 7: Classification of landslide-inducing factors used in this study (only the revised part is shown).**

| Predisposing Factor | Classification Criteria | Code |
|---|---|---|
| Slope Structure | Nearly horizontal slope | k |
| | Over-dip slope | |
| | Under-dip slope | |
| | Dip-oblique slope | |
| | Transverse slope | |
| | Anticlinal-oblique slope | |
| | Anticlinal slope | |

*6. Comment: In Table 7, the units of some landslide susceptibility factors are given, but the units for factors such as road density are missing.*

**Response**: Thank you for bringing this issue to our attention. We apologize for the omission of units for road density and two other landslide susceptibility factors. In the revised manuscript, we have added these missing units (line 287, in red font).

**Table 7: Classification of landslide-inducing factors used in this study (only the revised part is shown).**

| Predisposing Factor | Classification Criteria | Code |
|---|:---:|:---:|
| Road Density (km/km$^2$) | [0,0.5] | |
| | (0.5,1.2] | |
| | (1.2,2.5] | d |
| | (2.5,5.0] | |
| | >5.0 | |
| Tectonic Density (km/km$^2$) | [0,0.03] | |
| | (0.03,0.12] | |
| | (0.12,0.24] | f |
| | (0.24,0.38] | |
| | >0.38 | |
| Curvature (m$^{-1}$) | ≤-3 | |
| | (-3,-1] | |
| | (-1,0] | i |
| | (0,1] | |
| | >1 | |

*7. **Comment**: There are many images in the article. Consider combining some of them for display.*

**Response**: Thank you for your suggestion. Due to the extensive content of the article, the manuscript contains a large number of images and tables, which has resulted in excessive length, with some figures conveying limited information. Therefore, we have integrated the Thiessen polygon results from the original Figure 5 into Figure 2 (previously Figure 3), and combined the bar chart of historical disaster points from the original Figure 6 with Figure 3 (previously Figure 4). Additionally, the original Figure 11, which did not present meaningful information, has been removed. The revised content can be found in lines 138-139 and 152-154 (in red font).

[Figure]

Figure 2: Geographic location of the study area and Thiessen polygon results for rainfall stations.

[Figure]

Figure 3: Zoning map of the study area. (a) Schematic diagram of the sub-region merger; (b)

Number of historical landslide hazard sites in each sub-region.

**8. Comment**: *Some descriptions of the figures, such as the explanation of different colors in Figure 9 in lines 223-224, should be moved from the main text to the figure captions.*

**Response**: Thank you for pointing this out. We apologize for placing some explanatory notes that should have been in the figure or table captions within the main text, making the text overly cluttered. In response, we have reviewed and revised the explanatory text for all figures and tables in the manuscript. The revised figures and tables include Figure 5 (previously Figure 8), Figure 6 (previously Figure 9), Table 9, Figure 13 (previously Figure 17), and Figure 15 (previously Figure 19). The revised content can be found in lines 192-197, 209-213, 315-317, 329-332, and 352-355 (in red font).

[Figure]

**Figure 5: E-D rainfall threshold model results plotted using MLP regression. In the figure, regions are labelled as follows: a represents the Z11 region, b represents the Z12 region, c represents the Z13 region, d represents the Z21Z22 region, e represents the Z23Z24Z3 region, f represents the Z25Z4 region, and g represents the Dry Season. The red, blue, and purple points denote rainfall threshold values fitted for various landslide probabilities. Line segments are included solely for visual clarity and do not convey any practical information.**

[Figure]

**Figure 6: Schematic diagram of the E-D-R rainfall threshold model illustrated using the OLS regression results from the Z13 region as an example. The green, yellow, and red boxes in the figure represent landslide probabilities corresponding to rainfall thresholds of <25%, 25-50%, and 50-75%, respectively.**

**Table 9: Superposition matrix of landslide susceptibility and rainfall warning levels. In the table, the numerical codes represent the following zones: 1 – Relatively stable zone, 2 – General**

**prevention zone, 3 – Secondary prevention zone, and 4 – Priority prevention zone.**

| Susceptibility
Rainfall Threshold Level | Very Low | Low | Moderate | High | Very High |
|---|---|---|---|---|---|
| Caution | 1 | 1 | 1 | 1 | 2 |
| Special Caution | 1 | 1 | 1 | 2 | 3 |
| Warning | 1 | 1 | 2 | 3 | 4 |
| Severe Warning | 1 | 2 | 3 | 4 | 4 |

[Figure]

**Figure 13: Various rainfall parameters and rainfall warning levels for July 19, 2020. (a) Effective rainfall interpolated by Kriging; (b) Daily rainfall interpolated by Kriging; (c) Duration of rainfall estimated using Thiessen polygons; (d) Rainfall warning levels calculated using the optimal rainfall threshold model.**

[Figure]

**Figure 15: Transition process of rainfall warning levels in the Z12 region. The green line indicates the boundary between the Special Attention and Attention levels, the yellow line denotes the boundary between the Warning and Special Attention levels, and the orange line marks the boundary between the Severe Warning and Warning levels.**

*9. **Comment**: The clarity of Figure 14 is insufficient. It is recommended to change the layout from three columns to two columns.*

**Response**: Thank you for bringing this issue to our attention. Following your suggestion, we have changed the layout of the figure to two columns. The revised content can be found in lines 288-293 (in red font).

[Figure]

**Figure 10-1: Grading results for landslide-inducing factors. (a) Elevation; (b) NDVI; (c) TWI; (d) Road density; (e) Stratigraphic lithology; (f) Tectonic density.**

[Figure]

**Figure 10-2: Grading results for landslide-inducing factors (continued). (g) River distance; (h) Slope; (i) Curvature; (j) Land cover; (k) Slope structure.**

*Special thanks to you for your insightful and valuable comments in detail.*

---

## Author Comment (AC3)

**Responses to Reviewer:**

*General comments: In the MS titled "Optimizing Rainfall-Triggered Landslide Thresholds to Warning Daily Landslide Hazard in Three Gorges Reservoir Area", the authors tried to propose a rainfall threshold for predicting landslides on a daily scale. The topic fits the journal's scope while the entire MS was poorly structured, and the methods were not clearly explained. Hence a major revision is suggested.*

**Response**: We sincerely thank you for your recommendation and valuable comments, which have greatly contributed to improving this manuscript. We deeply appreciate the thorough and thoughtful review you have provided. In response to your comments, we have made detailed corrections, and we hope these revisions meet with your approval.

**Point by point responses to the nine comments**:

*1. Comment: There were too many abbreviations, making the MS hard to follow. Please reduce them to a reasonable number (less than 10).*

**Response**: Thank you for pointing this out. We fully agree with your suggestion that the excessive use of abbreviations made the manuscript more difficult to follow. In response, we have removed abbreviations like "rainfall threshold model (RTM)" and "landslide hazard warning (LHW)," retaining only well-known abbreviations such as RF, SVM, and MLP. We hope this adjustment will enhance the readability of the manuscript.

*2. **Comment**: The historical landslides were divided into 2 groups. For those that occurred during the dry season (41), "only rainfall thresholds for dry season landslides were calculated for the entire study area". The authors should explain what are the contributing factors for those dry season landslides? What kind of role of these factors are playing during the rainy season?*

**Response**: Thank you for bringing this to our attention. First, please allow us to explain the reasoning behind dividing the landslides into dry and rainy seasons: factors such as rainfall amount, soil moisture content, and vegetation cover vary significantly between the dry and rainy seasons, leading to substantial differences in rainfall thresholds. For this reason, we categorized the landslides into dry and rainy seasons. The further subdivision of the rainy season was done to better account for the potential impact of topographical factors on rainfall thresholds. In contrast, the small number of historical landslide events during the dry season made it impractical to further divide this group into subregions.

We also fully agree with your point that the influence of different factors on landslides can vary between the dry and rainy seasons, and that these influences should ideally be analyzed separately. However, the primary objective of this study is to investigate the differences in rainfall thresholds under varying topographical and climatic conditions and to validate the feasibility of real-time landslide hazard warning for the Three Gorges Reservoir area based on these thresholds. Due to limitations in manuscript

length and the scope of the research, we did not conduct separate assessments of landslide susceptibility during the dry and rainy seasons. We have added an explanation for this decision in Section 5.2. The revised content can be found in lines 384-387 (in red font). We hope our explanation will meet with your understanding.

It is also important to note that the spatial probability of landslide occurrence may vary between dry and rainy seasons, and the influence of different landslide-inducing factors may change under varying climatic conditions. This study primarily focused on the differences in rainfall thresholds across various climatic and topographic conditions, while the variations in spatial probability of landslide occurrence were not extensively explored.

*3. Comment*: *Following the above question, the water level fluctuation, and the underground water level might be important factors. But these factors had not been considered in section 4.2.1 or Table 5 "Source of data on landslide inducing factors".*

**Response**: Thank you for pointing this out. Indeed, reservoir water levels and groundwater levels are important factors in triggering landslides. However, due to the large scope of the study area, it was challenging to obtain comprehensive data on these factors, and therefore, they were not considered in this study. This limitation has been acknowledged in Section 5.2, where we have provided an explanation. The revised content can be found in lines

387-389 (in red font).

Additionally, changes in reservoir water levels and groundwater fluctuations in the Three Gorges Reservoir Area are significant factors influencing landslide occurrence; however, these factors were not included in this study due to data limitations.

*4. Comment: In Table 5, the human engineering activities were indicated using "road density", which seems not reasonable, unless it can be clearly figured out from the landslide inventory.*

**Response**: Thank you very much for your suggestion. The paper titled "Influence of human activity on landslide susceptibility development in the Three Gorges area" indicates that road construction is one of the most significant human activities affecting landslide occurrence in the Three Gorges Reservoir area. Additionally, since most roads in mountainous areas are constructed on cut slopes, their impact range is difficult to standardize. The paper "Prolonged influence of urbanization on landslide susceptibility" (Rohan et al., 2023) used road density to differentiate between urban and non-urban areas, effectively addressing the challenge of accurately determining road impact ranges. Inspired by this approach, we adopted road density instead of proximity to roads as a factor representing human engineering activities that may trigger landslides. In the revised manuscript, we have added references to relevant literature to support our findings. The revised content can be found in lines 253 and 542-543 (in red font).

Based on the research findings of previous scholars (Chen et al., 2021; Chen et al., 2020; Habumugisha et al., 2022; Li et al., 2022; Li et al., 2020; Rohan et al., 2023) and considering the specific conditions of the study area, this study selected a total of 11 factors that potentially induce landslides. These factors include elevation, Normalized Difference Vegetation Index (NDVI), Topographic Wetness Index (TWI), road density, stratigraphic lithology, tectonic density, river distance, slope, curvature, land cover, and slope structure (Table 5).

Li, Y.W., Wang, X.M., Mao, H., 2020. Influence of human activity on landslide susceptibility development in the Three Gorges area. Nat. Hazards 104, 2115-2151.

**5. Comment**: *The methodology and the framework should be elaborated using a figure. It reads confusing as it includes too many results (Figs. 7-20, Tables 1-8) using several methods from machine learning to threshold curve yielding in different zones.*

**Response**: Thank you for bringing this to our attention. We understand your concern regarding the complexity of the content and the lack of a flowchart. To more clearly illustrate the methods and steps involved in the article, we have added a flowchart in the revised manuscript (lines 72-74, in red font).

The study flowchart is shown in Fig. 1.

[Figure]

**Figure 1. Flowchart of this study.**

**6. Comment**: *Some of the figures are useless, such as Figures 1-2. Some of the figures should be combined.*

**Response**: Thank you for your suggestion. MLP and CNN are important models in machine learning and have been widely applied in various research fields in recent years. However, the description of the MLP and CNN frameworks in this paper took up considerable space. Additionally, due to the extensive content of the article, the manuscript contained numerous figures and tables, resulting in excessive length, with some figures conveying limited information. Therefore, in the revised manuscript, we have deleted the original Figures 1 and 2 and removed some of the foundational descriptions. Furthermore, we integrated the Thiessen polygon results from the original Figure 5 into the current Figure 2, and combined the bar chart of historical disaster points from the original Figure 6 with the current Figure 3. The

original Figure 11, which did not present meaningful information, was also deleted. The revised content can be found in lines 138-139 and 152-154 (in red font).

[Figure]

**Figure 2: Geographic location of the study area and Thiessen polygon results for rainfall stations.**

[Figure]

**Figure 3: Zoning map of the study area. (a) Schematic diagram of the sub-region merger; (b) Number of historical landslide hazard sites in each sub-region.**

*7. Comment: Try to find a fault map and include it in Figure 3, it's important and should not be ignored in this mountainous area.*

**Response**: Thank you for bringing this to our attention. We agree with your suggestion that fault distribution is crucial for studying landslide hazards. In the revised manuscript, we have added fault data to the current Figure 2 (lines 138-139, in red font).

[Figure]

**Figure 2: Geographic location of the study area and Thiessen polygon results for rainfall stations.**

*8. Comment: The details of the rainfall data should be introduced, including the covering period, the temporal resolution, etc.*

**Response**: Thank you for pointing this out. Two types of rainfall data were used in the study, and we apologize for not clarifying this earlier. In the revised manuscript, we have added details regarding the temporal and spatial

resolution of the forecasted rainfall data and distinguished it from the rainfall station data used in the landslide cataloguing. The revised content can be found in lines 221-223 (in red font).

Notably, the rainfall forecast stations used here were established later and differ from the rainfall stations used in the landslide cataloguing (Fig. 2, Rainfall Station). These forecast stations, covering the entire study area at 0.05° intervals, provide real-time updates on forecasted rainfall.

*9. Comment*: *The Thiessen polygon method was adopted to delineate the study area and the rainfall station (Figure 5), but it is not convincing. The zonation of the rainfall was also conducted (Figure 4). Why two methods were applied for one factor?*

**Response**: Thank you very much for your suggestion. We apologize for the lack of clarity in our original explanation, which led to some misunderstanding. When cataloguing landslides, it was necessary to obtain rainfall data for each historical landslide event from five days before the landslide occurred to the day of the event. Therefore, we used the Thiessen polygon method to delineate the rainfall station areas to identify which station corresponded to each landslide event and obtain the relevant rainfall data. The subsequent subregion delineation was performed to account for potential differences in rainfall thresholds under varying climatic and topographic conditions. This delineation was aimed at enhancing the accuracy of rainfall

thresholds within smaller regions. In the revised manuscript, we have adjusted the logical structure and rewritten section 3.2 to avoid any ambiguity. The revised content can be found in lines 140-163 (in red font).

**3.2 Landslide Data Cataloguing and Study Area Subdivision**

Cataloging landslide data is crucial for studying rainfall thresholds (Gariano et al., 2021). This process involves recording essential information, including the time of occurrence, geographic location, and associated rainfall stations for each landslide event. The historical landslide data used in this study were provided by the Wuhan Geological Survey Center (http://www.wuhan.cgs.gov.cn/). To identify the corresponding rainfall stations for each historical landslide, the Thiessen polygon method was employed to match each landslide point with the nearest rainfall station (Zhao et al., 2019), thereby obtaining the pre-landslide rainfall data (see Fig. 2, Thiessen polygons).

After filtering and cleaning, a total of 453 historical landslides with accurate rainfall information, dates, and locations were identified (see Fig. 2, Landslides). Historical rainfall data indicate that precipitation in the study area is primarily concentrated between May and October. The differing climatic conditions between the dry and rainy seasons may lead to varying impacts of rainfall on landslide movements (Soralump et al., 2021). Based on this information, the historical landslides were classified into rainy season and dry season landslides according to their occurrence times (Fig. 3(b)).

Given the substantial influence of geomorphological, geological, and climatic

conditions on landslide triggers during the rainy season (Dahal and Hasegawa, 2008), rainfall thresholds can vary across different regions. Accordingly, this study further subdivided the landslide data from the rainy season. The study area was divided into several sub-regions based on terrain and climatic conditions, with rainfall thresholds calculated for each region. However, due to the limited historical landslide data in regions $Z_{21}$, $Z_{22}$, $Z_{23}$, $Z_3$ and $Z_4$, adjacent regions were merged to mitigate potential inaccuracies in rainfall threshold calculations caused by insufficient data. Specifically, $Z_{21}$ and $Z_{22}$ were combined; $Z_{23}$, $Z_{24}$, and $Z_3$ were combined; and $Z_{25}$ and $Z_4$ were combined. The final regional subdivision is illustrated in Fig. 3(a). For dry season landslides, due to relatively uniform rainfall and the small number of events, no further subdivision was performed, and the rainfall threshold was calculated for the entire study area.

**10. Comment**: *In Figure 8, as the landslide data is not sufficient, the rainfall threshold results are derived using scattered E-D points. So, why do the authors have to conduct the zonation of rainfall stations?*

**Response**: Thank you for pointing this out. As we mentioned in our response to Comment 9, the use of the Thiessen polygon method to delineate the rainfall station areas was intended to accurately obtain the rainfall data for each historical landslide event from five days before the landslide occurred to the day of the event. Regarding the issue of insufficient landslide data points, the calculation of rainfall thresholds requires precise information

about the occurrence time and location of historical landslide events, as well as the corresponding rainfall data for those periods. After data cleaning and processing, only 453 historical landslide data points were available for use. To account for spatial variability as much as possible, we adopted a method of partially merging subregions to explore the optimal rainfall thresholds. The results indicate that in regions with a higher number of historical landslide data points, more accurate results were obtained. However, in regions with fewer historical landslide data points, the rainfall thresholds derived from the MLP method were indeed less accurate.

We addressed this issue in Section 5.3, where we highlighted the significant impact of the insufficient number of historical landslide data points on the certainty of rainfall thresholds. Additionally, we emphasized the need to establish a comprehensive historical landslide database. As new landslide events occur, the rainfall thresholds for the relevant subregions can be recalculated. As historical landslide data accumulate, the accuracy of the rainfall thresholds will continuously improve and become more stable.

**11. Comment**: *Figure 9 shows the E-D-R threshold model for a specific zone, that is, $Z_{13}$. Why only $Z_{13}$?*

**Response**: Thank you very much for your comment. The E-D-R threshold model considers three dimensions, resulting in different rainfall warning levels that are nested within each other in three-dimensional space. To

provide a clearer visualization of the E-D-R threshold model results, we used the Z13 region as an example and created the corresponding figure. The E and R axis values in the figure correspond to the rainfall thresholds obtained for the Z13 region using OLS regression. In the revised manuscript, we have clarified in the figure caption that this figure is provided as an example using the Z13 region, to eliminate any unnecessary ambiguity or misunderstanding. The revised content can be found in lines 209-212 (in red font).

[Figure]

**Figure 6: Schematic diagram of the E-D-R rainfall threshold model illustrated using the OLS regression results from the Z13 region as an example. The green, yellow, and red boxes in the figure represent landslide probabilities corresponding to rainfall thresholds of <25%, 25-50%, and 50-75%, respectively.**

*12. **Comment**: "Warning Daily Landslide Hazard" reads confusing. I guess the authors want to emphasize the warning was daily, but why?*

**Response**: Thank you for pointing this out. The core idea of this paper is to use rainfall thresholds combined with forecasted rainfall data to provide realtime warnings of landslide hazards in the Three Gorges Reservoir area. Due to the randomness and suddenness of landslide events, the risk level of landslides in the same region can vary at different times. Therefore, a single landslide hazard assessment may not be sufficient to support comprehensive prevention efforts, especially in a large area like the Three Gorges Reservoir Area. In this paper, we emphasize "daily" warnings, aiming to utilize a real-time updated rainfall forecasting system to obtain dynamically changing rainfall warning levels, thereby enabling daily updates to landslide hazard warnings. These daily updates allow relevant personnel to focus more precisely on high-risk areas, thus achieving low-cost, high-efficiency landslide disaster response. We hope our response has clarified your concerns regarding the term "Warning Daily Landslide Hazard." We greatly value the issue you raised and hope that our explanation has provided a clearer understanding of the intentions and methods behind our research.

**13. Comment**: *The writing should be significantly improved. There were too many grammars and typos. The terms should be defined accurately, for instance, "the third dimension indicator 'rainfall for the day' (R)".*

**Response**: Thank you for bringing this to our attention. We agree with your suggestion that the term "rainfall for the day" was not accurately expressed. In the revised manuscript, we have corrected it to "daily rainfall" to accurately represent the amount of rainfall on the day of the landslide occurrence.

Additionally, we referred to the article "Three ways ChatGPT helps me in my academic writing," published in Nature, which provided useful editing tips, and utilized the ChatGPT tool to conduct a thorough and precise revision of the manuscript. The revised content has been marked in red font in the manuscript.

*Special thanks to you for your insightful and valuable comments in detail.*